# Synchronous beta rhythms of frontoparietal networks support only behaviorally relevant representations

Evan G Antzoulatos[1,2]*, Earl K Miller[1]*

[1]The Picower Institute for Learning and Memory, Department of Brain and Cognitive Sciences, Massachusetts Institute of Technology, Cambridge, United States; [2]Center for Neuroscience, Department of Neurobiology, Physiology and Behavior, University of California at Davis, California, United States

**Abstract** Categorization has been associated with distributed networks of the primate brain, including the prefrontal cortex (PFC) and posterior parietal cortex (PPC). Although category-selective spiking in PFC and PPC has been established, the frequency-dependent dynamic interactions of frontoparietal networks are largely unexplored. We trained monkeys to perform a delayed-match-to-spatial-category task while recording spikes and local field potentials from the PFC and PPC with multiple electrodes. We found category-selective beta- and delta-band synchrony between and within the areas. However, in addition to the categories, delta synchrony and spiking activity also reflected irrelevant stimulus dimensions. By contrast, beta synchrony only conveyed information about the task-relevant categories. Further, category-selective PFC neurons were synchronized with PPC beta oscillations, while neurons that carried irrelevant information were not. These results suggest that long-range beta-band synchrony could act as a filter that only supports neural representations of the variables relevant to the task at hand.

*For correspondence: eantzoulatos@ucdavis.edu (EGA); ekmiller@mit.edu (EKM)

**Competing interests:** The authors declare that no competing interests exist.

## Introduction

Executive brain functions are closely linked with a network of cortical areas in the prefrontal and posterior parietal cortices. Human imaging studies have shown widespread frontal and parietal cortex activation during a broad range of cognitive demands (*Duncan, 2010*; *Fedorenko et al., 2013*). Likewise, there is an intermixing of neurons throughout frontal and parietal cortex that have neural correlates of different cognitive functions: attention, working memory, decision-making, rule-coding, and categorization (*Andersen and Cui, 2009*; *Buschman and Miller, 2007*; *Crowe et al., 2013*; *Freedman and Assad, 2006*; *Goodwin et al., 2012*; *Hayden and Pasternak, 2013*; *Hussar and Pasternak, 2009*; *Jacob and Nieder, 2014*; *Merchant et al., 2011*; *Rishel et al., 2013*; *Salazar et al., 2012*; *Vallentin et al., 2012*). These two regions not only have remarkable similarity in their patterns of neural activity, but they are also functionally interdependent. Deactivating one decreases neural activity in the other (*Chafee and Goldman-Rakic, 2000*) and the temporal dynamics of neural information in one can mirror that of the other (*Chafee and Goldman-Rakic, 1998*; *Crowe et al., 2013*; *Merchant et al., 2011*). Thus, it is becoming increasingly clear that an understanding of executive functions will depend on further insight into how the circuits within and between prefrontal and parietal cortices interact.

Insight can be gained by examining synchrony in the rhythms of their activity, which can reflect network properties (*Miller and Buschman, 2013*; *Roberts et al., 2013*; *Thut et al., 2012*). There is already evidence that frontoparietal rhythmic synchrony underlies at least two cognitive functions. Frequency-dependent increases in their synchrony have been seen during shifts of attention

**eLife digest** A brain that could store only exact experiences would bog us down with details. We have instead evolved to be able to detect the common elements in different experiences and group them into meaningful categories. This imbues the world with meaning. We can recognize and respond appropriately to objects, situations and expressions even if we have never encountered those exact examples before. Without this ability, experiences would be fragmented and unrelated. Things would seem strange and unfamiliar if they differed even trivially from previous examples. This situation describes many of the characteristics of neuropsychiatric disorders such as autism and schizophrenia.

Most studies have focused on how single brain areas or single neurons categorize experiences. The brain, however, is composed of many interacting networks of neurons that extend across several different areas of the brain. Repetitive rhythms, or waves, of electrical activity generated by the neurons seem to play a major role in network interactions. These rhythms are given different names depending on how rapidly they oscillate. In order for our brain to work successfully, these rhythms synchronize across the relevant brain areas.

Antzoulatos and Miller trained monkeys to categorize stimuli as "Above" or "Below" depending on where dots and lines appeared on a screen. The activity of the neurons in two regions of the monkey's brain – called the prefrontal cortex and the parietal cortex – was recorded as each monkey performed the task. The recordings revealed that synchronized rhythms between the prefrontal and parietal cortices supported the monkey's ability to categorize the stimuli. Changes in how strongly the rhythmic electrical activity of the neurons was synchronized – particularly for a type of wave called a beta wave – conveyed information about the category of stimuli (i.e., whether they counted as Above or Below). Single neurons also conveyed this categorization, but unlike rhythms, they also carried irrelevant information. Therefore the synchronized beta waves could act as a filter for the features of an object or experience that are relevant to the task at hand.

The prefrontal cortex and the parietal cortex are only two of many brain areas involved in categorization. Much more territory remains to explore.

(*Buschman and Miller, 2007*) and during visual working memory (*Salazar et al., 2012*), functions long associated with both regions. More recently, single-neuron studies indicate that both areas also make a major contribution to visual categorization (*Antzoulatos and Miller, 2011*; *Cromer et al., 2010*; *Crowe et al., 2013*; *Freedman et al., 2001*; *Freedman and Assad, 2006*; *Goodwin et al., 2012*; *Rishel et al., 2013*; *Roy et al., 2010*; *Swaminathan and Freedman, 2012*; *Tsutsui et al., 2016*). Individual neurons in both areas carve up different types of sensory information (i.e., motion, shape, location) according to learned category boundaries. Further, the category information carried by neuron spiking in one area evolves in lock-step with the other (*Crowe et al., 2013*; *Goodwin et al., 2012*). Yet, rhythmic synchrony during visual categorization has not been examined.

Thus, we recorded local field potentials (LFPs) from both regions while monkeys made decisions about abstract spatial categories. We used a delayed-match-to-category task in which monkeys were required to categorize visual cues along an abstract spatial dimension and then make a match/ non-match decision about whether subsequent test stimuli belonged in the same spatial category as the sample cue. This type of match/non-match category decision has been shown to engage PFC and PPC neurons (*Cromer et al., 2010*; *Crowe et al., 2013*; *Freedman et al., 2001*; *Goodwin et al., 2012*; *Roy et al., 2010*; *Swaminathan and Freedman, 2012*). We found increases in synchrony between and within frontoparietal networks that depended on whether monkeys categorized a stimulus as 'Above' or 'Below'. Moreover, we found that spiking activity that reflected the category distinction was better synchronized to frontoparietal LFP category networks than spiking that encoded information irrelevant to the task.

## Results

### Monkeys' behavior showed the hallmarks of categorization

Two rhesus macaque monkeys were trained to perform a Go/No-go category matching task. The trial began when they held a bar and achieved fixation of a central target (*Figure 1A*). The initial Above vs. Below category boundaries were indicated by two horizontal lines on each side of fixation, which divided the top from bottom halves of the screen. After the monkeys initiated the trial, the boundaries disappeared and a sample cue appeared briefly at a location chosen randomly from a set of 16 locations (± a small jitter) on the right or left side of the screen (*Figure 1B*; 144 possible locations in total). After a 500 ms post-sample delay, the two horizontal lines reappeared at shifted positions. On half of the trials, the lines shifted clockwise (the left shifted up and the right shifted down by 4 degrees). On the other half of the trials, they shifted counter clockwise (the opposite). After an additional 1 s delay, a test stimulus appeared randomly at one of 624 possible locations (*Figure 1B*). This test stimulus was a category match if it appeared on the same side of the (shifted) horizontal line (Above or Below) as the sample had appeared relative to the original horizontal line, irrespective of its exact location. If the test stimulus was a match, the monkey had to release the bar within 1 s (Go) to receive a juice reward. If the test was a non-match, the animals continued holding the bar (No-go) until another test stimulus (always a match) appeared on the screen. Note that the category of the initial sample cue did not change with the boundary shift. The boundary shift only changed which test stimulus locations would qualify as a category match or non-match. Thus, the purpose of the boundary shift was to ensure that the monkeys were encoding an abstract category representation, one that was not solely based on retinotopic coordinates. The monkeys also could not make a decision based solely on the proximity between sample and test stimuli because they could often be quite far apart (e.g., on the right and left of the screen; see *Figure 1A*), yet still belong to the same category. Both monkeys performed the task quite well, with 92.77% of all trials (excluding fixation breaks) correct.

To further quantify the monkeys' behavior, we calculated the probability of categorizing a sample as Above. This ($P_{ABOVE}$) was defined as the probability of a Go response to a test stimulus above the shifted boundary lines or a No-go response to test stimuli below the shifted lines (the probability of categorizing a sample as Below was complementary, $P_{BELOW} = 1 - P_{ABOVE}$). This allowed us to quantify the monkeys' category judgments as a function of the exact location of the sample cue. *Figure 1C* indicates that the monkeys' categorization of Above (or Below) was flat across the wide range of sample locations (top). Overall, $P_{ABOVE}$ was at least 0.83 if the sample was also Above (above the horizontal meridian) and at most 0.13 if the sample was actually Below.

We further demonstrated the monkeys' ability to generalize within the same category and distinguish between the categories in several ways. First, because the sample and test stimuli could appear in separate visual hemifields, yet still belong in the same category, we computed probability of the monkeys mistakenly categorizing the sample and test as being in different categories if they appeared in opposite hemifields and categorizing them as being from the same category just because they appeared in the same hemifield (i.e., categorization based on Left vs. Right). Thus, just as we did for Above vs. Below, we computed the probability of the monkeys categorizing the sample as Left. This ranged from 0.39 to 0.57 (*Figure 1C*), which is what we would expect if the monkeys were disregarding Left vs. Right (irrelevant dimension) and categorizing by Above vs. Below (relevant dimension). Thus, the monkeys were generalizing across diverse locations within the same category and not mistakenly categorizing the sample and test as being in different categories simply because they were far apart.

Likewise, Above vs. Below category performance remained high (at least 85% correct), regardless of sample cue eccentricity, sample distance from the horizontal and vertical meridians, the visual quadrant or hemifield of sample display, and the distance between the sample and test stimuli (*Figure 1—figure supplement 1*). Finally, performance remained high (at least 85% correct) on trials in which the retinotopic location that was occupied by the sample stimulus changed categories after the shift in boundaries and that location became a category non-match (*Figure 1—figure supplement 1*). Thus, the monkeys' behavior showed the hallmark of categorization: sharp discrimination across the boundary and flat generalization on either side of a boundary. The monkeys maintained a category representation that was uncoupled from the particular spatial location of the sample stimulus.

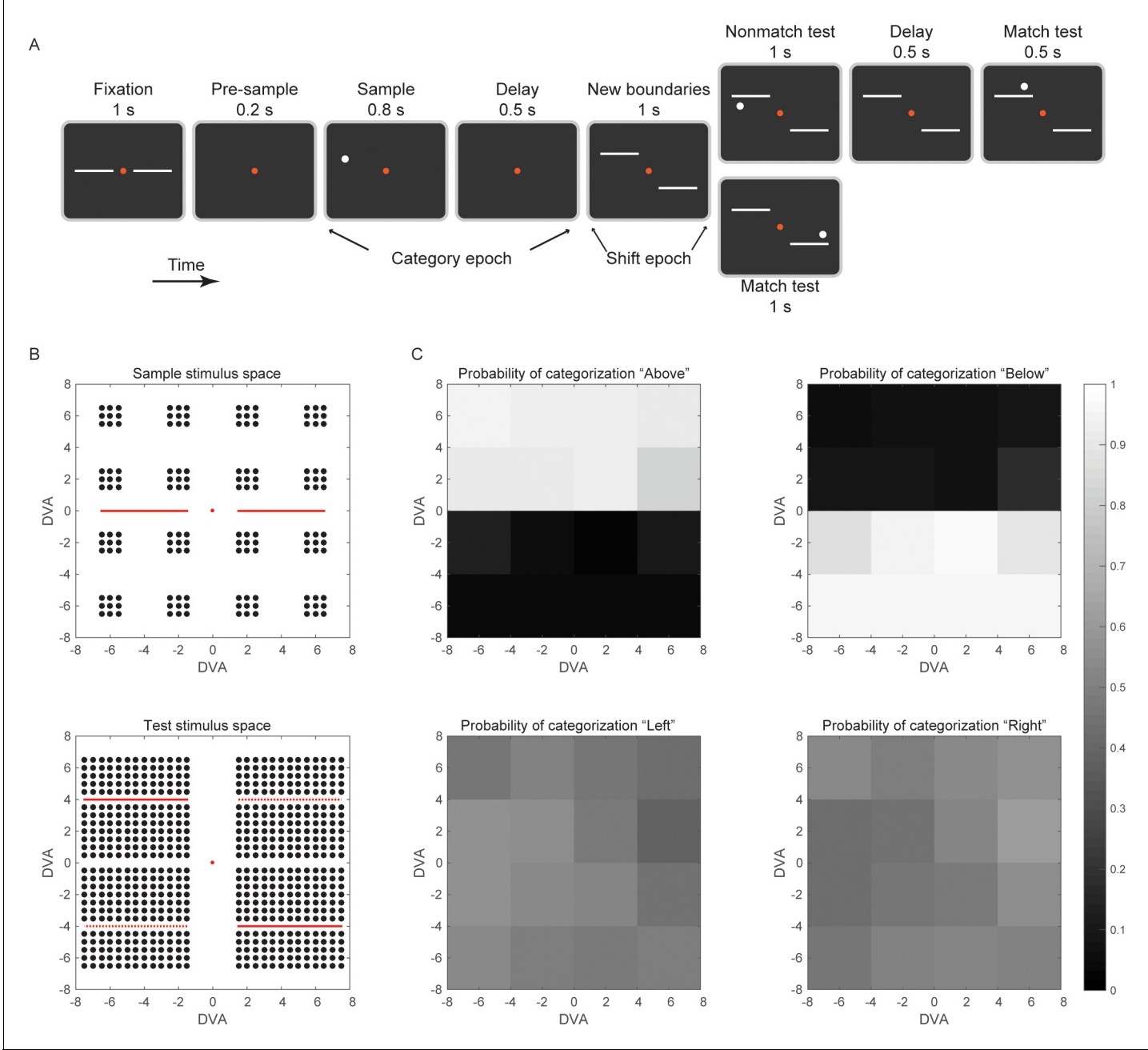

**Figure 1.** The task design. (**A**) The trial started when the animal held a bar and maintained visual fixation on the red central dot for 1 s. During fixation, in order to help the animal re-calibrate the Above/Below categorization boundary after the last trial, the 2 hemi-boundaries were also displayed. The sample stimulus appeared either above or below the horizontal meridian. To reduce the possibility of perceptual binding between the sample stimulus and the horizontal boundaries, there was a brief, 200 ms time gap between the two displays (pre-sample epoch). After the post-sample delay epoch, 2 hemi-boundaries were displayed at new locations, according to a CW (as shown) or CCW shift. The animal had to adjust the decision criterion based on the location of the new boundaries. When a test stimulus appeared, it had to be compared to the sample stimulus in a boundary-referenced, rather than a retinotopic, spatial frame. If a non-match, the first test would be followed by a second test, always a match to the sample category. Until display of a match test, the animal had to maintain visual fixation and contact with the bar. Upon a match test display, the animal had to release the bar for liquid reward. (**B**) Sample stimuli could appear randomly at any one of the 144 locations shown (top), with equal probability in the category Above or Below (DVA: degrees of visual angle). Test stimuli (equal probability for categories above and below the new boundaries) could appear at any one of the 624 locations shown (bottom). The continuous red horizontal lines in the test space indicate the position of hemi-boundaries after CW shift, and the dashed lines (dashed only for illustration) indicate the CCW shift. Each trial would test one of 89,856 possible sample-test combinations, equally distributed between match and non-match types. After the boundary shift, the Above/Below categories spanned a visual area that overlapped with the corresponding area of the pre-shift categories by 50%. (**C**) The probability to categorize a sample stimulus as 'Above', 'Below', 'Left', or 'Right', by

*Figure 1 continued on next page*

*Figure 1 continued*

spatial location. The animals would choose Left or Right with equal probability, because the Left/Right dimension was task-irrelevant. See also **Figure 1—figure supplement 1**.

The following figure supplement is available for figure 1:

**Figure supplement 1.** Performance on the categorization task.

## Predominant beta rhythms in local and long-range oscillations of the frontoparietal network

We recorded from multiple electrodes simultaneously advanced in the lateral PFC (lPFC, areas 46 and 45), caudal PFC (cPFC, area 8A and the FEF), and the anterior intraparietal area (AIP; see **Figure 2—figure supplement 1**). The AIP subregion of the posterior parietal cortex was selected because it is strongly coupled to premotor areas and plays a role in visuospatial transformations that guide motor movements (**Verhoef et al., 2015**). Data from subareas of the lPFC (as well as subareas of the cPFC) were combined because they yielded similar results. To examine interactions between the areas, we analyzed simultaneously recorded LFPs from a total of 159 electrodes in AIP, 111 electrodes in cPFC, and 129 electrodes in lPFC. From the LFP we first removed stimulus-evoked components (to isolate the task-induced oscillations), and decomposed it in the frequency domain using the Morlet wavelet. The LFP was decomposed in six octaves (from 2–128 Hz), at a resolution of 0.1 octave. From the amplitude of the wave components we then computed the LFP time- and frequency-dependent power and normalized it to 1/frequency (in order to correct the power-law decay of the LFP power spectrum). Removing the LFP components that are phase-locked to the stimulus may conceal high-frequency rhythms time-locked to stimulus presentation, but it is necessary to avoid spurious synchrony due to simultaneous stimulus-evoked responses in the LFPs.

This analysis revealed elevated power of the 16–32 Hz octave (beta band) in all three areas (**Figure 2A**). Beta-band power was especially prominent at the end of the sample presentation and in the early part of the immediately following delay. This is around the time that we would expect the monkey to categorize the location of the sample. Beta power was also evident just before presentation of the test stimulus, after the boundaries switched.

To compare beta power across areas and time, we divided the trial into two main epochs. The Category epoch lasted from the sample display onset up to the boundary shift (time 0–1.3 s). This is presumably when the monkeys categorized the sample as Above or Below. The Shift epoch started after the boundary shift and lasted up to the test stimulus display (time 1.3–2.3 s). At this time, the monkeys had to maintain in working memory the category of the sample despite the fact that the retinotopic location at which the sample had appeared might have become a category non-match after the boundary shift.

A 2-way ANOVA compared beta-band power among the three areas and two trial epochs. As suggested by **Figure 2A**, there was significantly greater power during the Category epoch (average: 0.06 mV$^2$ ± SEM: 0.002) than the Shift epoch (0.04 ± 0.002; p=$4.7 \times 10^{-10}$). The strongest beta power appeared in AIP, followed by cPFC (p=$3.4 \times 10^{-23}$; posthoc comparisons: AIP (0.065 ± 0.002) > cPFC (0.05 ± 0.002), p=$7 \times 10^{-10}$; cPFC > lPFC (0.03 ± 0.002), p=$2.6 \times 10^{-3}$; AIP>lPFC, p=$4.7 \times 10^{-12}$).

We next compared LFP synchrony within and between areas using the pairwise phase consistency metric (PPC). Synchrony was bias-corrected by subtracting the chance-level PPC, as estimated from a surrogate dataset. For every pair of electrodes, the surrogate dataset was created by randomly shuffling the trials of each electrode independently of the other electrode 200 times without replacement. The PPC that was computed across these random permutations was averaged and provided the baseline-level synchrony that would be expected by chance. The pattern of synchrony largely paralleled the spectral power results. Beta band synchrony was evident within the areas (**Figure 2B**; AIP-AIP: n = 340 pairs of electrodes; cPFC-cPFC: n = 205 pairs; lPFC-lPFC: n = 290 pairs) and between areas (**Figure 2C**; AIP-cPFC: n = 474 pairs; cPFC-lPFC: n = 323 pairs; AIP-lPFC: n = 535 pairs). A 2-way ANOVA compared synchrony as a function of area pairing and trial epoch. Like the power analysis, there was significantly greater beta synchrony in the Category epoch (0.09 ± 0.002) than the Shift epoch (0.07 ± 0.002; p=$2.7 \times 10^{-10}$). There were also significant differences in synchrony

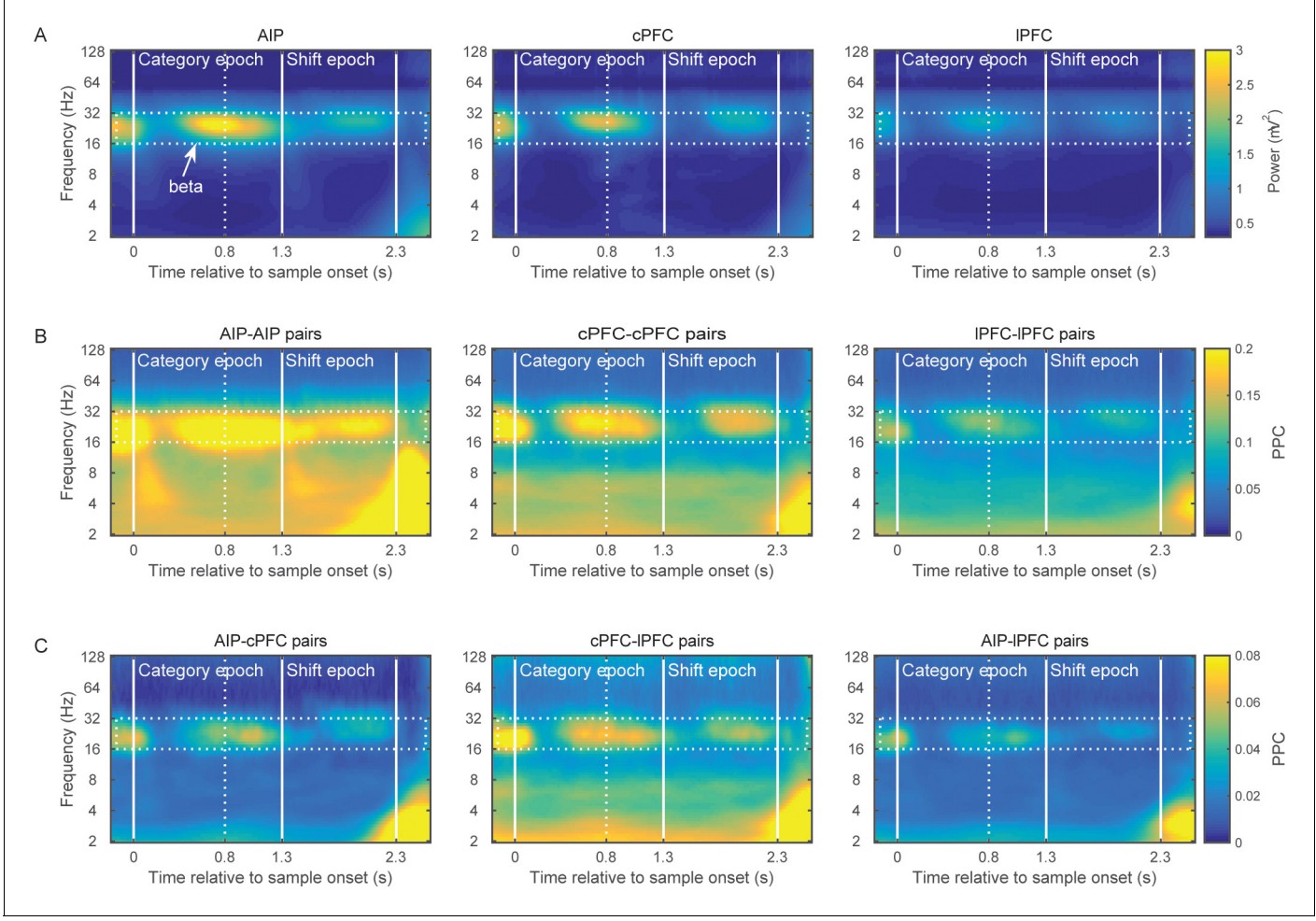

**Figure 2.** Brain rhythms of the frontoparietal network. (**A**) Average power (mV$^2$) of the LFPs recorded from AIP (left), cPFC (middle), and lPFC (right) electrodes in time-frequency space. Strong oscillations are seen at the beta band (16–32 Hz; dotted rectangle in all panels) in all three areas. For these analyses, the evoked component has been removed from the LFPs, and the power has been normalized to 1/f. Spectral analyses were performed at six octaves (2–128 Hz), at a 0.1 octave resolution (i.e., 10 frequency bins per octave). Vertical white lines demarcate the Category and Shift epochs, and dashed white lines indicate the end of sample display. (**B**) Strong synchrony (Pairwise Phase Consistency) of beta-band oscillations (16–32 Hz) is observed across pairs of electrodes within AIP (left), cPFC (middle) and lPFC (right). (**C**) As also seen with power (**A**) and intrinsic synchrony in the three areas (**B**), strong synchrony of beta-band rhythms is also seen in the extrinsic pairs of electrodes across the three areas: AIP-cPFC (left), cPFC-lPFC (middle), and AIP-lPFC (right). Recording sites appear in *Figure 2—figure supplement 1*. The corresponding figure before removal of the evoked component appears in *Figure 2—figure supplement 3* and the cross-area synchrony for match vs. non-match trials appears in *Figure 2—figure supplement 2*.

The following figure supplements are available for figure 2:

**Figure supplement 1.** Electrophysiological recording sites.

**Figure supplement 2.** Match vs.non-match trials.

**Figure supplement 3.** Power and synchrony of LFPs before removal of the evoked component.

between areas. The strongest was within AIP and weakest between AIP and lPFC. (Posthoc comparisons, corrected p<0.01: AIP-AIP (0.18 ± 0.003) > cPFC-cPFC (0.13 ± 0.004) > lPFC-lPFC (0.08 ± 0.003) > cPFC-lPFC (0.04 ± 0.003) > AIP-lPFC (0.02 ± 0.003); AIP-cPFC (0.03 ± 0.003) not different from AIP-lPFC and cPFC-lPFC).

The low-frequency synchrony that was observed late in the trial, during the test stimulus (*Figure 2B and C*), may be related to the motor response to a category match (in 50% of the trials), the reward that followed correct trials, or the eye movement that typically follows the release from visual fixation requirement. *Figure 2—figure supplement 2* illustrates the cross-area synchrony during the Shift and Test epochs separately for match and non-match trials, indicating that the strong low-frequency synchrony coincided with the animals' response.

## Category-selective synchrony

Previous studies have shown rule and category-selective beta synchrony within the PFC and between the PFC and striatum (*Antzoulatos and Miller, 2014*; *Buschman et al., 2012*), wherein different pairs of electrodes showed increased beta synchrony for one or the other category/rule. We found the same results for frontoparietal synchrony and spatial categories.

An example of category-selective synchrony of beta oscillations between the frontal and parietal cortex from one recording session is shown in *Figure 3*. It shows the filtered beta synchrony between a cPFC electrode (F09) and two different AIP electrodes (P07 and P10) for trials of category Above vs. Below. The cPFC (F09) electrode had stronger beta synchrony with one AIP electrode (P07) for Above (red and black traces, left) but stronger beta synchrony with the other AIP electrode (P10) for Below (red and green traces, right).

We quantified the degree of LFP synchrony for Above vs. Below trials using pairwise phase consistency (PPC), a measure of synchrony that is corrected for small and unequal sets of trials (*Vinck et al., 2010*). The absolute difference in PPC between the trials of the two categories was transformed into a z-score based on 200 random permutations of the trials between the two categories.

This revealed category-selective synchrony for Above/Below categories both between and within the areas (*Figure 4A and C* respectively), predominantly in two octaves: 16–32 Hz (beta), and 2–4 Hz (delta). The contours (black lines) in *Figure 4* indicate regions of time/frequency space where category selectivity was significantly greater than zero (p<0.05 at each bin for at least 60 ms and three consecutive frequencies). Category-selective synchrony between AIP and cPFC was mostly concentrated in the beta band during the Category epoch and in the delta band during the Shift epoch (*Figure 4A* left panel). Category selective synchrony between cPFC and lPFC electrodes (*Figure 4A* middle panel) was similar. By contrast, lPFC and AIP category selective synchrony was mostly in the low-frequency oscillations (*Figure 4A* right panel). Within AIP, category-selective synchrony was longer-lasting and covered a wider range of frequencies (*Figure 4C*). A similar pattern of results was found using the percentage of electrode pairs with category selective synchrony as a measure, instead of the average strength (*Figure 4—figure supplement 1*). In addition, the observed category-selective synchrony did not arise from a difference in the pattern of network dynamics between the categories. Both the synchrony between and within the frontal and parietal areas displayed similar time-frequency dependence for the category Above as for the Below (*Figure 4—figure supplement 2*).

We examined whether the category selectivity of frontoparietal synchrony arose from a selective loss or gain of synchrony for the preferred/non-preferred categories relative to baseline. To address this, we averaged the synchrony observed in the beta and delta bands during the Category and Shift epochs, separately for trials of the Above and Below categories. Every electrode pair's category preference was determined from the polarity of the difference Above-Below. For every pair of frontoparietal electrodes we compared the epoch- and frequency-specific synchrony to the corresponding synchrony observed during the fixation epoch (the last 750 ms of the fixation window), separately for the preferred and non-preferred categories. *Figure 4—figure supplement 3* shows results only for the AIP-cPFC pairs of electrodes because they showed overall stronger category-selective synchrony than the AIP-lPFC pairs (*Figure 4*). The directional changes from baseline were mixed (although all were statistically significant), but the trend was for the beta-band synchrony during the Category epoch (i.e., the epoch that indicated stronger selectivity in beta synchrony; *Figure 4*) to increase from baseline for the preferred category and to decrease from baseline for the non-preferred category. The same trend was observed for the delta-band synchrony during the Shift epoch (the epoch with stronger selectivity of delta synchrony). In the other two epochs (when synchrony selectivity was weaker; *Figure 4*) synchrony of both the beta and delta rhythms was

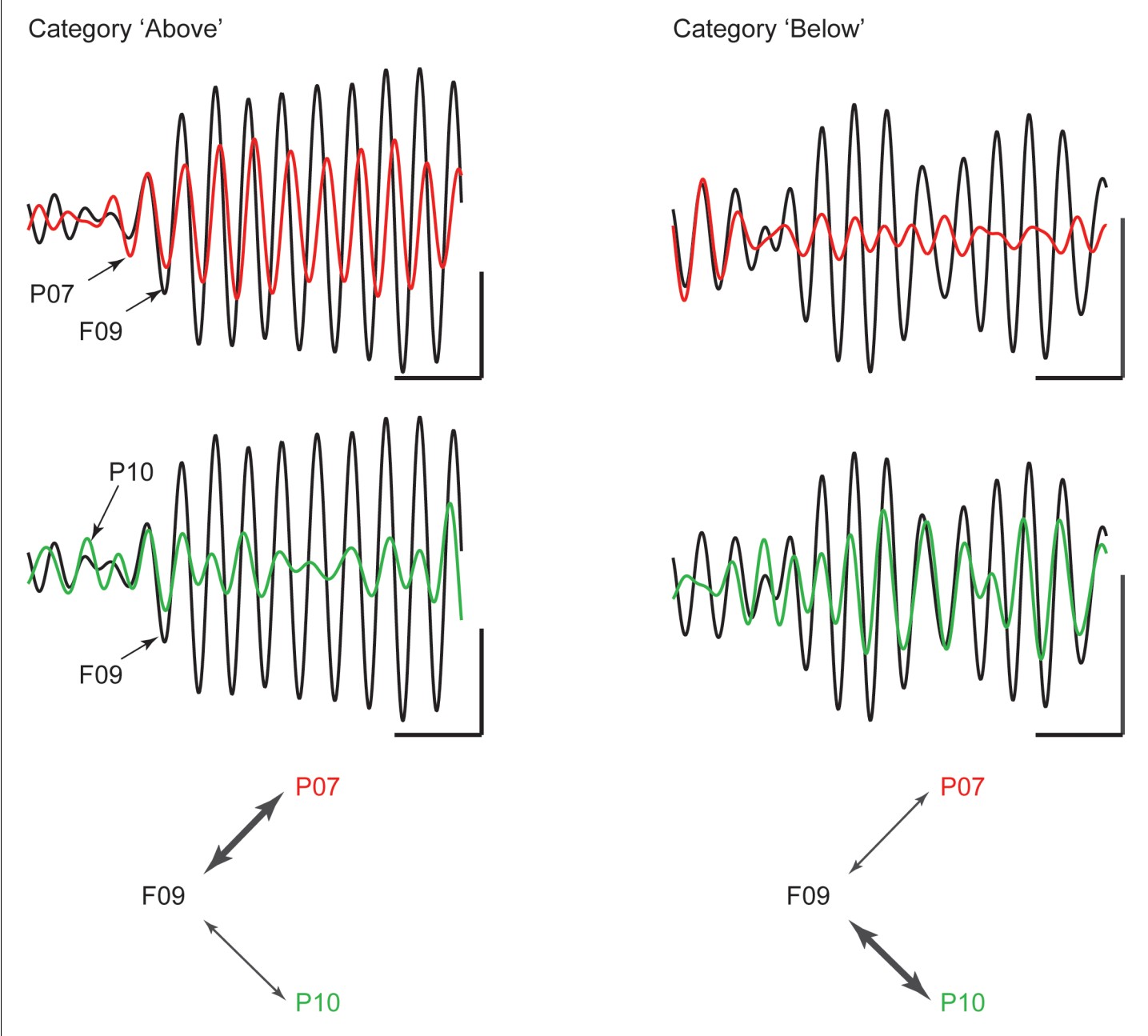

**Figure 3.** Example traces of beta rhythms with category-selective synchrony. During a trial of category Above (left), a cPFC electrode (F09; black trace) displays stronger phase and amplitude coupling with one AIP electrode (P07; red) than with another (P10; green). The strength of coupling among the same electrodes is reversed during a trial of category Below (right). The cPFC electrode now couples more weakly with the P07 than with the P10 electrode. Scale bars: 0.4 mV and 100 ms.

decreased from baseline, albeit to a smaller extent for their preferred than for their non-preferred category.

It was possible that the category-selective synchrony described above did not reflect the Above vs. Below categories per se but, instead, differences in the retinotopic location of the sample cue that were unrelated to the categorization task. If so, we would expect to see differences in synchrony between sample cues appearing in the left vs. right hemifields. After all, receptive fields in both the frontal and parietal cortex are strongly biased to the contralateral field. However, when we re-sorted

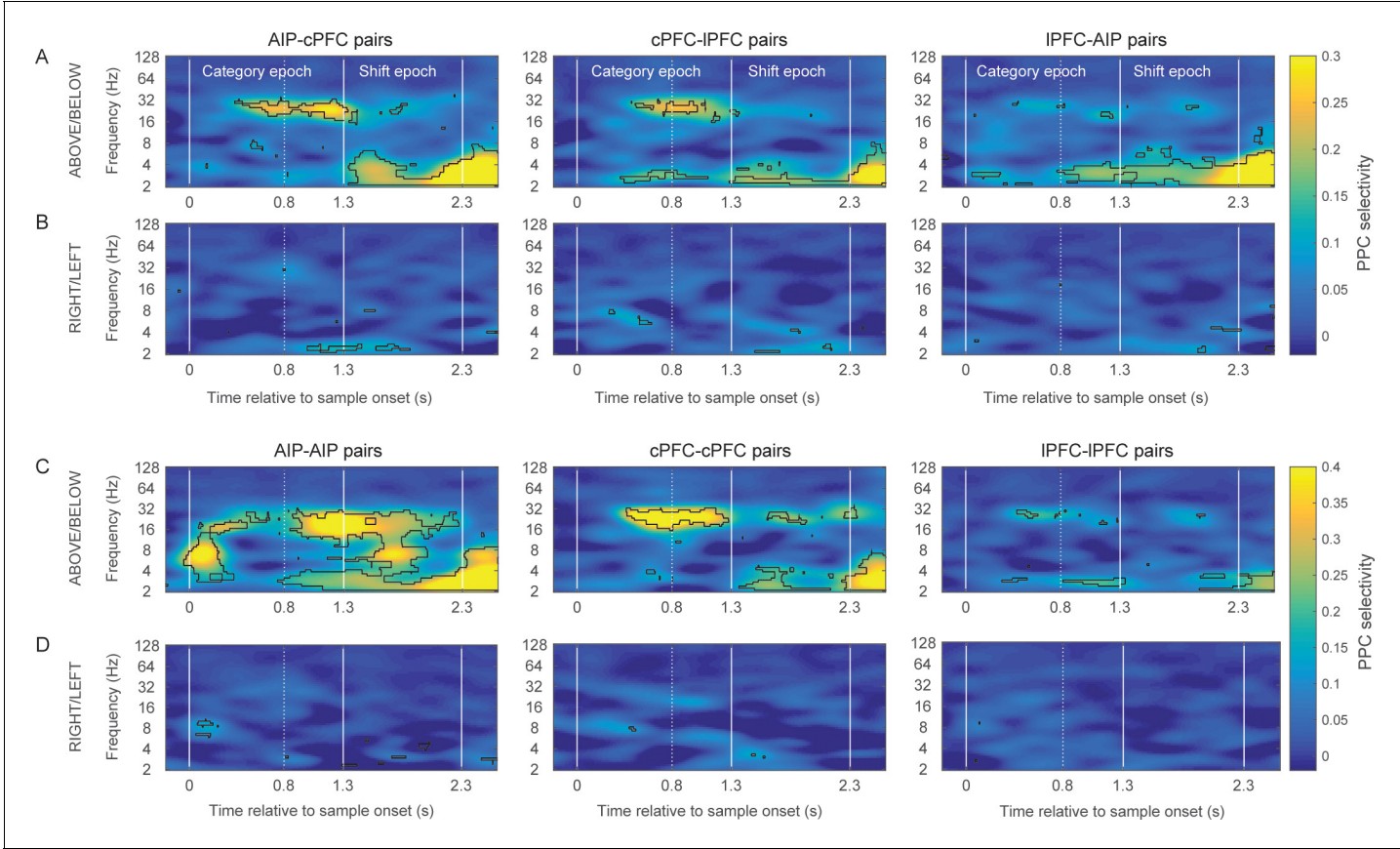

**Figure 4.** Category-selective synchrony of frontoparietal rhythms. (**A**) and (**B**) Population average of PPC selectivity of all simultaneously recorded pairs of AIP and cPFC (left), cPFC and lPFC (middle), lPFC and AIP (right) electrodes along the task-relevant, Above/Below, dimension (**A**) and task-irrelevant, Right/Left, dimension (**B**). (**C**) and (**D**) PPC selectivity of intrinsic pairs of electrodes within AIP (left), cPFC (middle) and lPFC (right) along the Above/Below (**C**) and Right/Left (**D**) dimensions. In all panels, contours identify regions of time-frequency space with statistically significant population selectivity. Similar time-frequency dependent patterns were seen in the percent of sites with significantly selective synchrony (see *Figure 4—figure supplement 1*). The time-frequency dynamics of PPC were similar for both the Above and Below categories (see *Figure 4—figure supplement 2*).

The following figure supplements are available for figure 4:

**Figure supplement 1.** Prevalence of category-selective synchrony of frontoparietal rhythms.

**Figure supplement 2.** Category-specific synchrony of oscillations in the frontoparietal network.

**Figure supplement 3.** Frontoparietal synchrony for preferred/non-preferred categories vs. pre-trial baseline.

the trials based on whether the sample cue appeared in the left vs. right hemifields, there was virtually no selectivity in beta synchrony (*Figure 4B and C*). Thus, category-selective synchrony mirrored the demands of the task: To encode the Above vs. Below category of the sample, irrespective of Right vs. Left.

We collapsed the frequency and time dimensions by taking the average synchrony across each of the two trial epochs and two frequency bands, beta and delta. We performed a 4-way ANOVA on z-transformed PPC values (area x epoch x category boundary x frequency band), which included six area combinations (AIP-cPFC, cPFC-lPFC, lPFC-AIP, AIP-AIP, cPFC-cPFC, and lPFC-lPFC), two trial epochs (Category and Shift), two types of category boundaries (relevant: Above/Below and irrelevant: Right/Left), and two frequency bands (beta and delta). This analysis confirmed what was seen in the time-frequency analysis (*Figure 4*). It indicated significant differences among the areas ($p = 2.4 \times 10^{-25}$), significantly greater selectivity for the Above/Below than the Right/Left boundary

(p=1.97×10$^{-168}$), and significantly greater selectivity of beta synchrony during the Category epoch and of delta synchrony during the Shift epoch (band x epoch interaction: p=1.2×10$^{-32}$).

*Figure 5* illustrates in more detail the interactions among the experimental variables area, band, category boundary, and trial epoch, as estimated from the ANOVA post-hoc comparisons. It plots the mean synchrony of each group along with its 95% confidence interval (CI) after the Bonferroni correction for multiple comparisons. As seen at the upper half of *Figure 5*, both beta and delta rhythms displayed synchrony that was significantly selective for Above vs. Below (i.e., the corrected 95% CIs were greater than zero). None of the area combinations showed beta synchrony selective for Right vs. Left (bottom quadrant of *Figure 5*). Delta synchrony was not selective for Right vs. Left within any area but was selective in the cross-area interactions (i.e., AIP-cPFC, cPFC-lPFC, and AIP-lPFC) and only during the Shift epoch. Thus, pure category-selective synchrony was restricted to the beta band. There was more delta selectivity for Above vs. Below but also some delta selectivity for Right vs. Left.

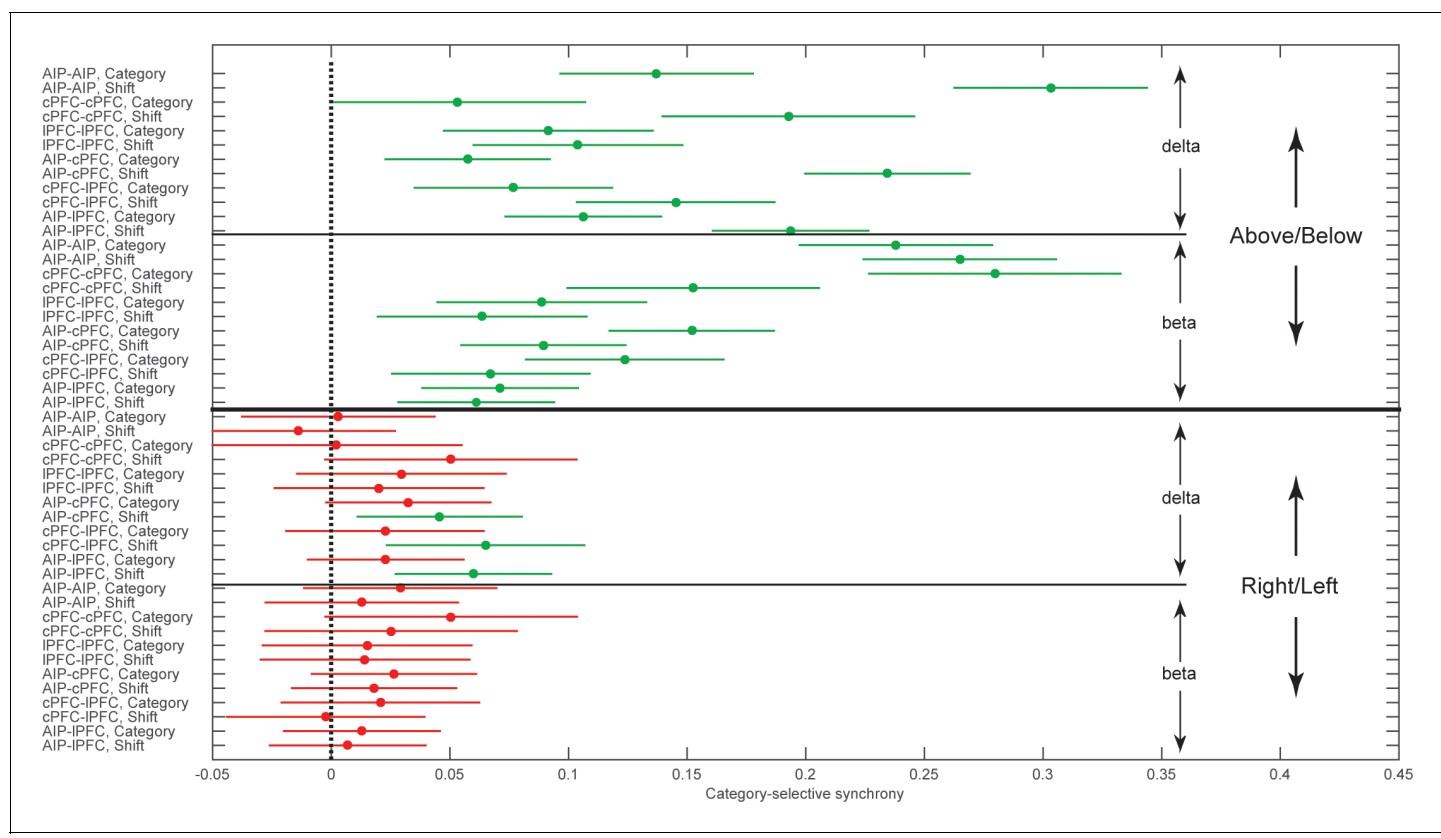

**Figure 5.** Group averages of category-selective synchrony of LFPs. Filled dots represent group averages and horizontal lines the 95% confidence intervals (CI) after correction for multiple comparisons. The upper half of the graph illustrates selectivity for the task-relevant category boundary (Above/Below) and the lower half the selectivity for the task-irrelevant, Right/Left boundary. Within each boundary, the upper halves show the category-selective synchrony of delta rhythms and the lower halves the selective synchrony of beta rhythms. Within each quadrant of the graph, the rows correspond to the 2 trial epochs (Category and Shift) for each of the six combinations of the areas studied. Groups with no overlapping confidence intervals are significantly different (e.g., the Above/Below selectivity of AIP-AIP delta in the Shift epoch is significantly greater than the selectivity of most other groups). Category selectivity is significantly greater than zero (dashed vertical line) in all groups with green markers, and not significantly different from zero in groups with red markers (i.e., depending on whether a CI includes zero). Synchrony of beta rhythms displays significant selectivity for the Above/Below but not for the Right/Left boundary. In turn, synchrony of delta rhythms displays significant Above/Below selectivity, while Right/Left selectivity is area and epoch-dependent. The data of the figure can also be seen in *Figure 5—source data 1* .

The following source data is available for figure 5:

**Source data 1.** Group averages (and SEM) for the post-hoc comparisons plotted in *Figure 5*.

## Comparison to spiking activity

From the same electrodes we isolated the spiking activity of 1078 single neurons (296 from cPFC, 387 from lPFC, and 395 from AIP). As previously reported (*Goodwin et al., 2012*), spiking activity in these areas showed selectivity for the learned spatial categories (Above vs. Below). This is illustrated in *Figure 6A*, which shows the average selectivity (assessed as percent explained variance or PEV) across the entire population of all isolated neurons from each of the three areas as a function of time. The colored bars at the top of the figure show when the PEV values of each area were significantly greater than zero (t-test, corrected p<0.05). Note that selectivity for the Above vs. Below categories was evident shortly after the onset of the sample cue and continued throughout the remainder of the trial, growing stronger in the delay following the sample. This figure also shows that category selectivity in spiking activity was strongest in cPFC, next in lPFC, and weakest in AIP (effect of area from 2-way ANOVA: $p=7.9\times10^{-22}$; post-hoc comparisons with corrected p<0.05: cPFC (1.24 ± 0.07) > lPFC (0.62 ± 0.06) > AIP (0.39 ± 0.06)). *Figure 6B* shows similar results with an alternative measure, i.e. the percentage of neurons in each area that showed a significant effect of category as a function of time.

For our LFP analyses (discussed above), the beta band was of particular interest because beta oscillations have been associated with a variety of cognitive functions and previous studies have shown category and rule selectivity in this band (*Antzoulatos and Miller, 2014*; *Buschman et al., 2012*). As noted above, there was category-selective beta-band synchrony within and between each area. However, there was virtually no selective beta synchrony for the (task-irrelevant) Right vs. Left hemifield location of the sample cue. By contrast, the spiking of single neurons was sensitive to the Right vs. Left location of the sample. This is shown in *Figure 6C and D*. The average population activity of all three areas showed significant selectivity for Right vs. Left early on during sample display, albeit weaker than the learned Above vs. Below categories. Right vs. Left selectivity was strongest in the prefrontal cortex (cPFC and lPFC) and weak in AIP (effect of area from 2-way ANOVA: $p=3.1\times10^{-8}$; post-hoc comparisons with corrected p<0.05: AIP (0.1 ± 0.02) < cPFC (0.22 ± 0.02) and lPFC (0.27 ± 0.02), cPFC not different from lPFC). Thus, information about the task-irrelevant Right vs. Left location of the sample 'leaked through' in spiking activity but did not seem to do so for the beta synchrony (but, as noted, did for the delta synchrony). As shown in *Figure 6—figure supplement 1*, these analyses yielded similar results when performed at the level of population spiking activity (i.e., on the pooled spikes of all neurons isolated from each electrode, rather than the activity of each individual neuron).

Another notable difference between the patterns of results for synchrony vs. spike rate was in the strength of effects across areas. As noted above, beta power and synchrony was strongest within AIP and between AIP and the other areas. Beta category-selectivity was also strongest within AIP. However, as *Figure 6* shows, AIP showed the weakest category selectivity in the spiking rate of its neurons. This is not because there were fewer isolated neurons in AIP vs. other areas (see above for numbers of neurons from each area). The average number of isolated neurons per electrode was also similar across areas (AIP: 2.48 neurons/electrode; cPFC: 2.67; lPFC: 3; note that electrodes with zero neurons were not included in any of the LFP analyses). LFP synchrony displayed similar dynamics between pairs of electrodes that had the same category preference in their spiking activity as those that had different category preferences (*Figure 6—figure supplement 2*).

## Asymmetric frontoparietal interactions

The above results indicate category-selective patterns of LFP synchrony within and between the parietal and prefrontal cortices. Here, we examine synchrony between spikes in one area and LFPs in another. Synchrony between spikes in area A (the neuronal output from A) and LFPs in area B (presumably reflecting the synaptic inputs to B), but not the other way around, may suggest a unidirectional influence from A to B.

We started with pairs of electrodes from different areas that displayed Above/Below category-selective beta synchrony. For each pair of electrodes we first averaged the category selectivity (i.e., the data plotted in *Figure 4*) of LFP synchrony separately and the Category and Shift epochs. Then, we identified the electrode pairs that showed the strongest 10% of selectivity in beta synchrony in each epoch. We computed spike-field synchrony between them, using the same method that we previously used to examine frontostriatal interactions (*Antzoulatos and Miller, 2014*). A phase-

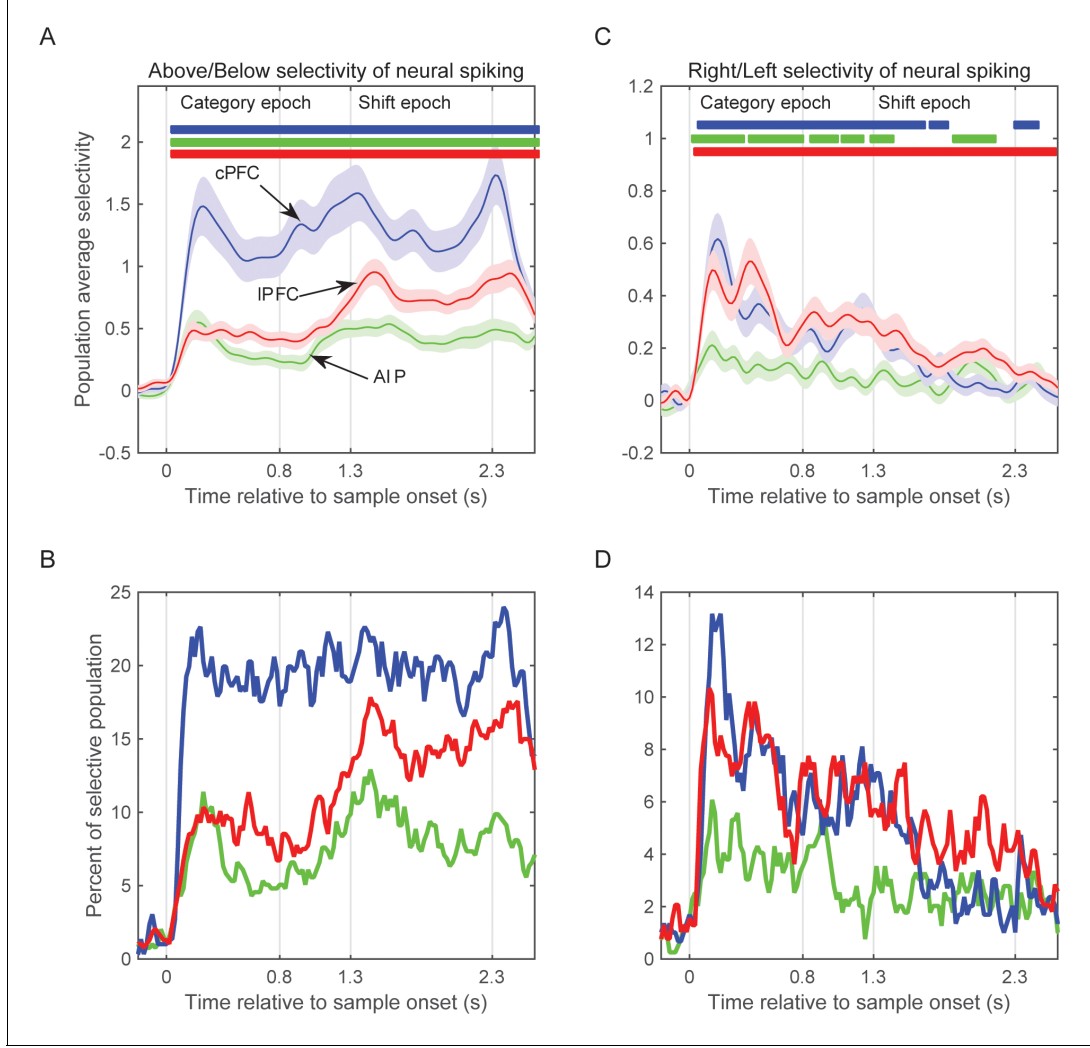

**Figure 6.** Selectivity of neural spiking activity. (**A**) Average selectivity (±SEM) of neural spiking for the task-relevant category of sample location (Above/ Below) in each of the three neural populations as a function of time from sample onset. Horizontal bars indicate time bins of significant selectivity (p<0.05 lasting for at least 100 ms) in the corresponding populations (blue: cPFC; green: AIP; red: lPFC). (**B**) Percent of each population with significant selectivity (p<0.01) for the Above/Below categorization. (**C**) Same as **A** for the task-irrelevant dimension of sample location (Right/Left). (**D**) Same as **B** for the Right/Left dimension. Overall, all three populations showed stronger selectivity for Above/Below than for Right/Left (note difference in y scales between left and right panels), and different time courses between Right/Left selectivity (which gradually decayed after sample display) and Above/ Below selectivity (which tended to grow stronger after the end of sample display). The results are similar if selectivity is analyzed on the pooled spikes from all neurons isolated from each electrode (see *Figure 6—figure supplement 1*). Synchrony of LFP oscillations between two electrodes showed similar selectivity whether the corresponding spiking had the same or different category preferences (see *Figure 6—figure supplement 2*).

The following figure supplements are available for figure 6:

**Figure supplement 1.** Selectivity of pooled multi-unit activity.

**Figure supplement 2.** LFP synchrony in relation to individual site preference.

locking value (PLV) between spikes and LFPs (see Materials and methods) indicated whether spikes in one area are more likely to occur at specific LFP phases from an electrode in other areas. Because this metric can show spurious synchrony if the number of spikes is small (*Antzoulatos and Miller, 2014*), we took two mitigating measures. First, we pooled together all the spikes from isolated neurons that were recorded from each electrode, thus maximizing the number of spikes in each electrode. Second, we transformed the PLV to a z score based on 200 random permutations of the LFP.

By keeping each trial's series of spikes and permuting only the LFP trace we were able to evaluate the levels of spike-LFP PLV that arose solely from the number and temporal pattern of the spikes. We first focused on the Category epoch because that is where we observed the strongest category-selective synchrony of beta oscillations (*Figure 4*).

This analysis revealed significant spike-LFP synchrony between cPFC spikes and AIP LFP beta oscillations. An example is shown in *Figure 7A*, where the red trace corresponds to the 25–35 Hz oscillations of an AIP LFP, and the vertical black bars indicate spike times simultaneously recorded from a cPFC electrode. As the superposition indicates, the cPFC spikes tended to appear at the same phase of the LFP oscillation, namely the downstroke of the LFP cycle. Average results from all

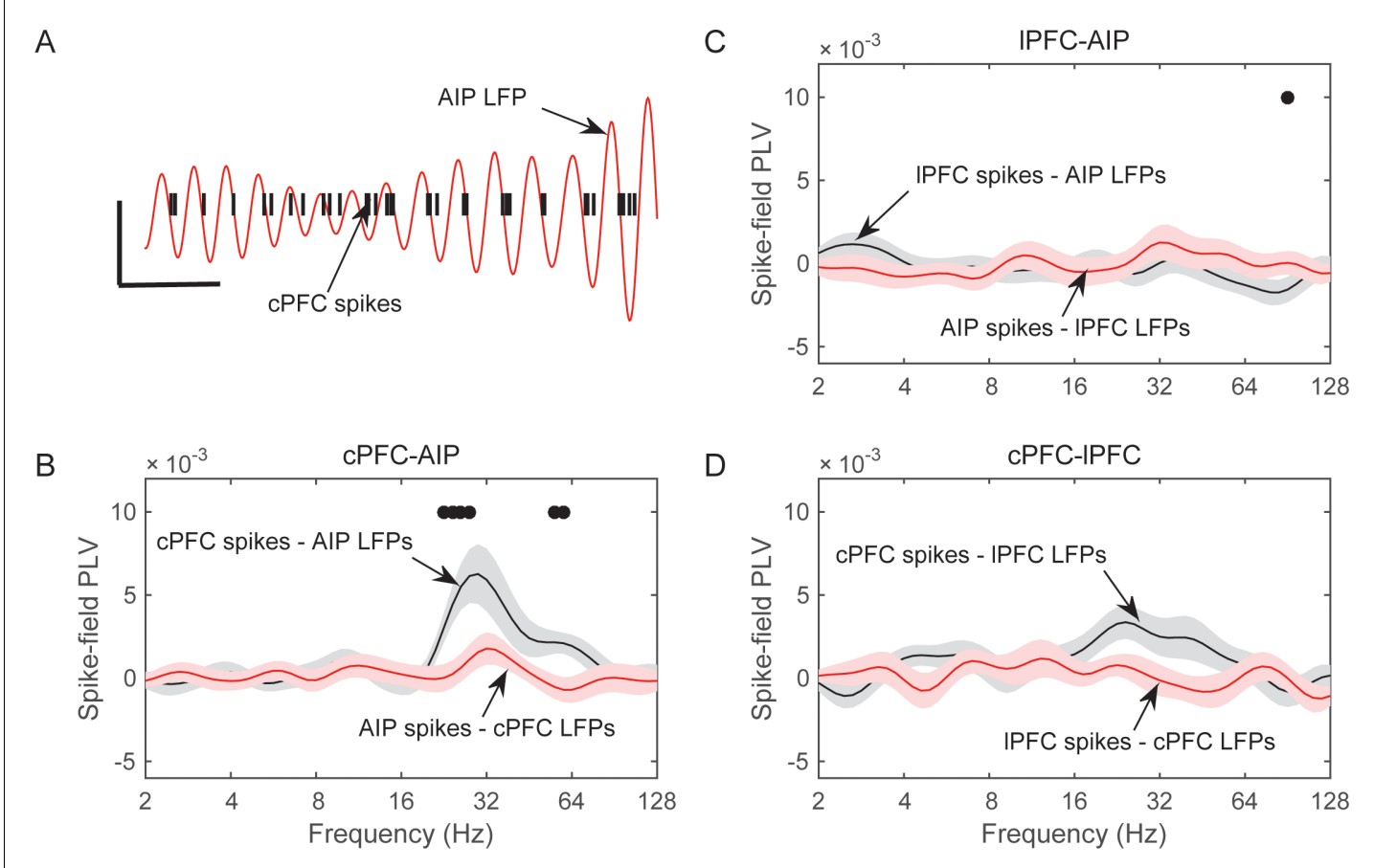

**Figure 7.** Frequency-dependent synchrony of spiking activity to LFPs during the Category epoch. (A) Example LFP (red trace; passband filtered at 25–35 Hz) from an AIP electrode, simultaneously recorded with spiking activity from a cPFC electrode (black bars). Prefrontal spikes are phase-locked to the downstroke of the parietal LFP. Scale bars: 100 ms and 0.2 mV. (B) Average (±SEM) synchrony (PLV) between spikes in cPFC and LFPs in AIP (black trace), or the reverse (red trace) as a function of frequency, for pairs of electrodes with category-selective synchrony of beta oscillations (*Figure 4*). Dots indicate significantly different spike-field synchrony (p<0.05) between the two directions. (C) Same as in B, for pairs of electrodes between lPFC and AIP. (D) Same as in B and C, for pairs of electrodes between cPFC and lPFC. See also *Figure 7—figure supplement 2* for spike-field synchrony during the Shift epoch, as well as spike-field synchrony between pairs of electrodes with category-selective delta synchrony. The asymmetry in cPFC-AIP vs. AIP-cPFC spike-LFP synchrony is stronger in the category Below than Above (see *Figure 7—figure supplement 1*).
The following figure supplements are available for figure 7:

**Figure supplement 1.** Category-specific spike-field synchrony between cPFC spikes and AIP LFPs.
**Figure supplement 2.** Spike-LFP synchrony.
**Figure supplement 3.** Category-selective synchrony between cPFC-AIP LFPs.

pairs of spikesand LFPs during the Category epoch are shown in panels B-D of *Figure 7* for each combination of areas (i.e., cPFC-AIP: n = 47 pairs, lPFC-AIP: n = 54, and cPFC-lPFC: n = 32). The dots indicate when the PLV between spikes in one area and LFPs in the other were significantly greater than the other way around (2-tailed t-test, p<0.05). As can be seen from this figure, significant spike-LFP synchrony was between cPFC spikes and AIP LFPs, with a peak at 30 Hz (*Figure 7B*). This synchrony was asymmetric: there was significantly stronger phase-coupling of cPFC spikes to AIP LFP oscillations in the 22–32 Hz band (0.006 ± 0.002) than the other way around (−0.001 ± 0.001; phase-coupling of cPFC oscillations to AIP spikes, t-test: p=0.004). This asymmetry was more pronounced for the category Below than Above (*Figure 7—figure supplement 1*). No frequency-dependent spike-LFP synchrony was observed between lPFC spikes and AIP LFPs or the reverse (*Figure 7C*). There was only modest synchrony between cPFC spikes and lPFC LFPs in the beta band, but this was not significantly unidirectional (*Figure 7D*).

There was only modest spike-LFP synchrony during the Shift epoch (*Figure 7—figure supplement 2*), with modest but significant directionality between the two prefrontal subregions (coupling of cPFC spikes to lPFC 12–16 Hz oscillations (0.002 ± 9.5×10$^{-4}$) was stronger than the reverse (−0.002 ± 9.01×10$^{-4}$), p=0.004). Finally, we performed the same analyses on select pairs of electrodes that showed strong category-selective synchrony in the delta band. We only saw modest synchrony between cPFC spikes and AIP beta oscillations (*Figure 7—figure supplement 2*). These results indicate that the pairs of frontoparietal sites that displayed category-selective beta synchrony also showed strong spike-LFP synchrony in the beta band. Conversely, pairs of frontoparietal sites that displayed strong spike-LFP synchrony displayed more pronounced category-selective synchrony of beta rhythms (*Figure 7—figure supplement 3*).

## Preferential spike-field synchrony of the task-relevant spikes

We next asked whether spike-LFP synchrony across areas (evaluated as above) was stronger when spiking activity carried task-relevant information. Indeed, we found that cPFC electrodes with spiking selectivity for the relevant category distinction (Above vs. Below) showed stronger synchrony with AIP beta oscillations than the cPFC electrodes with spiking selectivity for the irrelevant dimension (Right vs. Left).

From each neural population (cPFC, lPFC, and AIP) we first averaged the selectivity of each electrode's spiking activity within the Category and Shift epochs (i.e., the PEV metric shown in *Figure 6—figure supplement 1*). We then selected the electrodes at the top 10% of spiking selectivity for the Above/Below dimension and the electrodes at the top 10% for the Right/Left dimension (group statistics appear in *Table 1*). This resulted in the same number of electrodes for each dimension within each population (cPFC: n = 11 electrodes for Above/Below selectivity and same number but different electrodes for Right/Left selectivity; lPFC: n = 13; AIP n = 16). The selection of the top 10% was necessary because the Above/Below selectivity was more prevalent than the Right/Left selectivity (*Figure 6—figure supplement 1*, panels B and D). Selecting the top 10% eliminated this bias. We

**Table 1.** Electrodes with spiking at the top 10% of selectivity.

| Neural population | | | cPFC | lPFC | AIP |
|---|---|---|---|---|---|
| Above/Below selectivity | Category epoch | Mean | 11.74 | 3.57 | 3.96 |
| | | SEM | 2.14 | 0.49 | 0.30 |
| | Shift epoch | Mean | 9.12 | 6.54 | 4.53 |
| | | SEM | 1.31 | 0.75 | 0.43 |
| Right/Left selectivity | Category epoch | Mean | 2.69 | 3.26 | 1.64 |
| | | SEM | 0.41 | 0.42 | 0.21 |
| | Shift epoch | Mean | 1.30 | 1.93 | 0.88 |
| | | SEM | 0.26 | 0.31 | 0.11 |

Average (±SEM) selectivity during the Category and Shift epochs for the Above/Below and Right/Left dimensions of sample location is provided for each neural population of the analyses in *Figure 8*.

subsequently evaluated the synchrony of spikes from each of these electrodes with all the simultaneously recorded LFPs from other electrodes, regardless of the category selectivity in the synchrony between the corresponding pair of LFPs.

The results of the cPFC-AIP analyses appear in *Figure 8*. In the Category epoch, we found significantly stronger synchrony of AIP LFPs with cPFC spikes that had Above/Below selectivity than with cPFC spikes that had Right/Left selectivity (peak at 32 Hz, *Figure 8A*, t-test: p=9.3×10$^{-7}$). In fact, synchrony of cPFC spikes to AIP LFPs was significant for spikes with Above/Below selectivity (n = 54 spike-LFP pairs, 0.01 ± 0.002, t-test comparison to zero, p=1.7×10$^{-6}$), but not different from zero for spikes with Right/Left selectivity (n = 44 spike-LFP pairs, −0.003 ± 0.002, p=0.98). By contrast, synchrony between AIP spikes and cPFC LFPs was significant (*Figure 8B*; Above/Below: n = 50, 0.006 ± 0.002, p=3.7×10$^{-4}$; Right vs. Left: n = 52, 0.004 ± 0.001, p=0.001) but there was no difference between AIP spikes with Above/Below vs. Right/Left selectivity (p=0.32). The difference between cPFC spike rates with Above/Below selectivity vs. spike rates with Right/Left selectivity was similar for both the Above and Below categories (*Figure 8—figure supplement 1*). Similarly, spike rates from electrodes without any Above/Below selectivity (the bottom 10% of PEV) also displayed weaker spike-LFP beta synchrony than the electrodes with spike rates with strong Above/Below selectivity (*Figure 8—figure supplement 2*). By contrast, there was little or no cPFC-AIP spike-LFP synchrony in the Shift epoch for either Above/Below or Right/Left selective spikes (*Figure 8C,D*). Finally, there was only modest spike-LFP synchrony between lPFC and AIP, or between cPFC and lPFC, in either direction, regardless of Above/Below or Right/Left selectivity (*Figure 8—figure supplement 3*).

These results suggest a relationship between the magnitude of category selectivity a cPFC electrode had in its spike rate and the synchrony of that spiking to the parietal beta oscillations. Indeed, as *Figure 9* illustrates, there was significant linear (but not rank) correlation between the Above/Below spike rate selectivity of cPFC electrodes and their synchrony to the AIP beta rhythms (averaged across all simultaneously recorded AIP electrodes: Pearson's r=0.25, p=0.009; Spearman's rank r=0.16, p=0.09). By contrast, there was no linear (or rank) correlation between the Right/Left spike rate selectivity of the same cPFC electrodes and their synchrony to parietal beta rhythms (Pearson's r=−0.11, p=0.28; Spearman's rank r=−0.04, p=0.70).

## Discussion

We found that during a rule-based spatial categorization task, there was an increase in beta-band oscillatory power and synchrony within and between the prefrontal cortex and parietal cortex. Different pairs of recording sites, both within and between the prefrontal cortex and parietal cortex, showed category selectivity, that is, increased LFP synchrony for one or the other category, primarily in the beta but also in the delta band. Thus, each category was accompanied by different network patterns of synchrony. Prefrontal spikes were synchronized to parietal LFP beta rhythms, but parietal spikes were not synchronized to prefrontal rhythms. This suggests a unidirectional influence with the PFC influencing the parietal cortex. Finally, we found that prefrontal neural spiking selective for the task-relevant Above vs. Below category dimension was synchronized with the parietal beta rhythms, but spiking that reflected the task-irrelevant Right vs Left dimension was not.

Oscillatory synchrony has been suggested to play a role in establishing functional networks (*Engel et al., 2001*; *Fries, 2005*; *Miller and Buschman, 2013*; *Uhlhaas et al., 2009*). Our finding of category-selective patterns of increased synchrony between recording sites in the frontoparietal cortex supports this. It adds to previous reports of content-specific patterns of oscillatory synchrony in these circuits (and beyond). For example, there is object-selective synchrony between the prefrontal and parietal cortex for different objects held in working memory (*Salazar et al., 2012*) and within the prefrontal cortex for different behavioral rules (*Buschman et al., 2012*). Further, category-selective patterns of synchrony develop between the prefrontal cortex and striatum in parallel with category learning (*Antzoulatos and Miller, 2014*). These 'rhythmic ensembles' do not seem to be limited to circuits involving the prefrontal cortex. Selective synchrony of gamma rhythms between V4 and V1 has been reported to guide attentional selection in primates (*Bosman et al., 2012*). A recent study in humans suggests functional networks across the cortex are formed by harmonic patterns constrained by known anatomy (*Atasoy et al., 2016*). In other words, it seems as if anatomy (the connectome) may be the infrastructure that determines which functional circuits can potentially

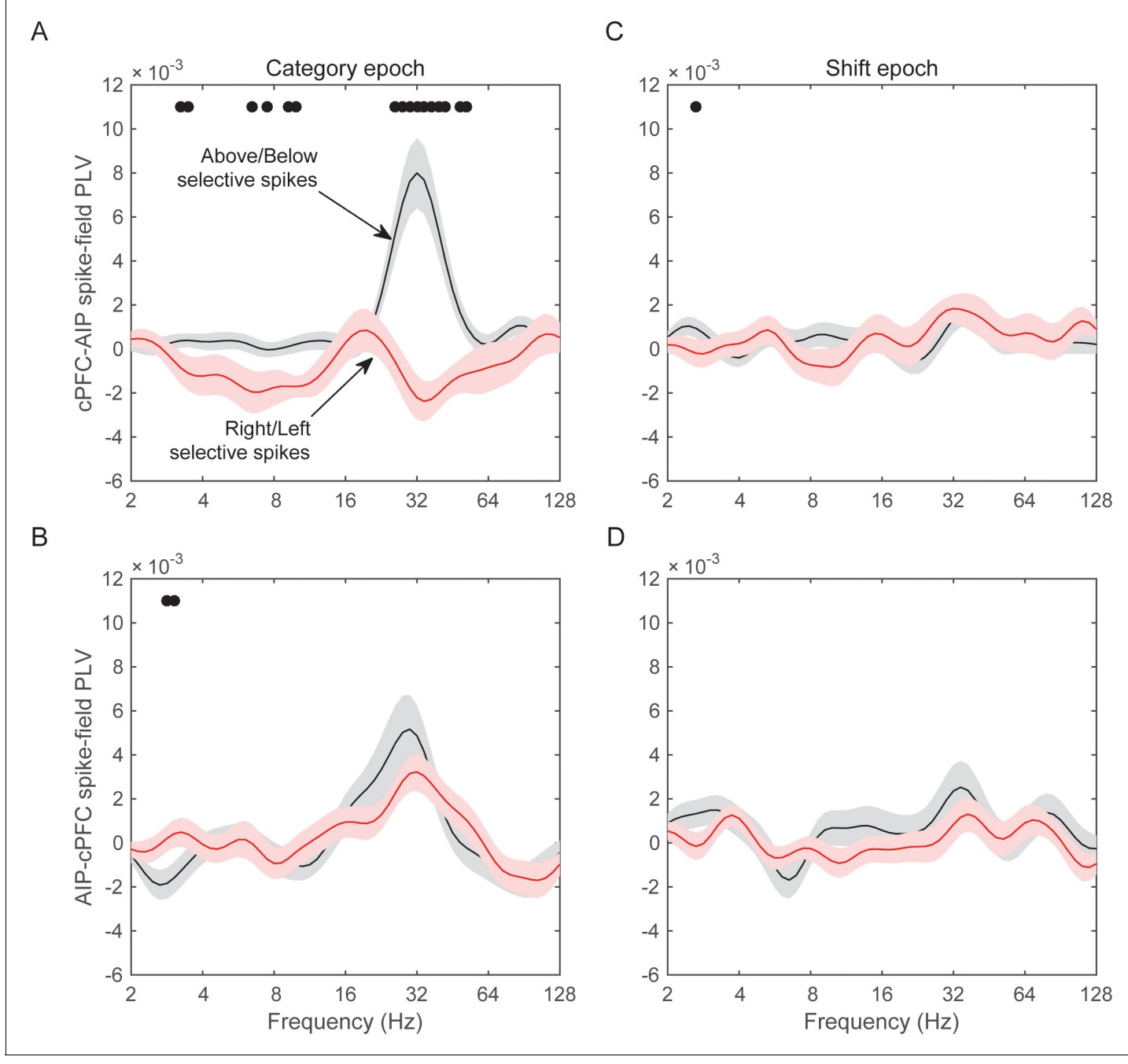

**Figure 8.** Spike-LFP synchrony for selective spiking. (**A**) Average (± SEM) PLV between cPFC spikes and AIP LFPs as a function of frequency, separately for electrodes with strong Above/Below selectivity in their spiking (black trace) and electrodes with strong Right/Left selectivity in their spiking (red trace) during the Category epoch of the categorization trials. Dots mark frequencies of significant difference between the red and black traces (2-tailed t-test; p<0.05). (**B**) Same as (**A**), but in reverse direction: synchrony between AIP spikes and cPFC LFPs. (**C**) and (**D**) Same as (**A**) and (**B**) respectively, but for the Shift epoch of the categorization trials. The group statistics of the electrodes used in these analyses (electrodes at the top 10% of spiking selectivity) appear in *Table 1*. See also *Figure 8—figure supplement 3* for the corresponding analyses on the other combinations of spike-LFPs. See *Figure 8—figure supplement 2* for a comparison of spikes with vs. without Above/Below selectivity. *Figure 8—figure supplement 1* illustrates cPFC-AIP spike-LFP synchrony separately for each category.

The following figure supplements are available for figure 8:

**Figure supplement 1.** Spike-field synchrony of cPFC-AIP pairs separated by category.

*Figure 8 continued on next page*

*Figure 8 continued*

**Figure supplement 2.** Spike-field synchrony for category-selective vs.non-selective spiking.
**Figure supplement 3.** Spike-field synchrony for selective spikes.

form. Which circuits do form from moment to moment may be regulated, at least in part, by patterns of synchrony. To use an analogy: Anatomy is the roads, activity is the traffic, and rhythmic synchrony is the traffic lights.

In prior reports, as well as in this study, these 'rhythmic ensembles' tend to center on the beta band (*Antzoulatos and Miller, 2014*; *Buschman et al., 2012*; *Haegens et al., 2011*; *Micheli et al., 2015*; *Salazar et al., 2012*; *Siegel et al., 2009*). This is noteworthy because there is increasing evidence that beta-band synchrony may play a role in sustained top-down, or feedback, signaling in the cortex (*Bastos et al., 2015*; *Engel and Fries, 2010*). In line with this is our observation that category selectivity was more 'pure' in synchrony (especially in the beta band) than for spiking activity. While beta synchrony was selective for the task relevant categories, we could not detect any (irrelevant) information about the left vs. right hemifield location of the sample cue in beta synchrony. By contrast, left vs. right was detectable in neural spike rates. Of course, selectivity of spiking rate and LFP synchrony are two different phenomena, which rely on different measures and analyses. Thus, this difference could merely reflect differences in the ability to detect information in them. But it is also possible that beta synchrony is more selective. Beta synchrony may provide an additional level of filtering beyond that seen in spiking activity. Because of its putative role in feedback processing in the cortex, it may only pass back top-down information to other cortical areas. Support for this also comes from our finding that only the prefrontal spiking activity that conveyed task-relevant information was synchronized to the parietal beta rhythms. By contrast, for pairs of recording sites with category-selective beta synchrony, we did not detect any synchrony between parietal spikes with prefrontal oscillations. This all suggests a top-down influence from the PFC to parietal cortex.

Our results are in line with other studies that indicate prefrontal and parietal cortex involvement in visual categorization (*Braunlich et al., 2015*; *Crowe et al., 2013*; *Goodwin et al., 2012*; *Swaminathan and Freedman, 2012*). Neurons in both areas show correlates of a variety of abstract cognitive factors such as number, shape and spatial categories (*Goodwin et al., 2012*; *Murata et al., 2000*; *Tudusciuc and Nieder, 2009*). Chafee and colleagues and Merchant and colleagues have previously reported single-neuron correlates of spatial categories. By analyzing the fluctuations of the time course of spiking from both areas, they also concluded that category information was transmitted in a top-down fashion from the PFC to parietal cortex (*Crowe et al., 2013*; *Merchant et al., 2011*).

In sum, our results suggest that the spatial categories may be computed in prefrontal circuits, where this information intermingles with bottom-up information (in this case, the task-irrelevant distinction between the hemifields). But the prefrontal cortex selectively broadcasts via beta rhythms, and thus strengthens, only the relevant information back to the parietal cortex. Long-range fronto-parietal networks supported by synchronized beta rhythms, then, permit the most relevant representation to dominate and guide the animal's category decision.

## Materials and methods

### Animals

Data were collected from two adult male rhesus macaque monkeys (*Macaca mulatta*), weighing 10–12 kg. The animals were taken care of in accordance with the National Institutes of Health guidelines and the policies of the Massachusetts Institute of Technology Committee for Animal Care. Both animals were trained on the rule-based categorization task until they reached similar levels of proficiency. Each of them was surgically implanted with one titanium headbolt and two titanium recording chambers, one of which was placed above the left principal sulcus and the other above the left intraparietal sulcus. Placement of the chambers as well as estimation of the electrode recording sites were guided by structural MRI scans and computed with Matlab (Mathworks, Natick MA).

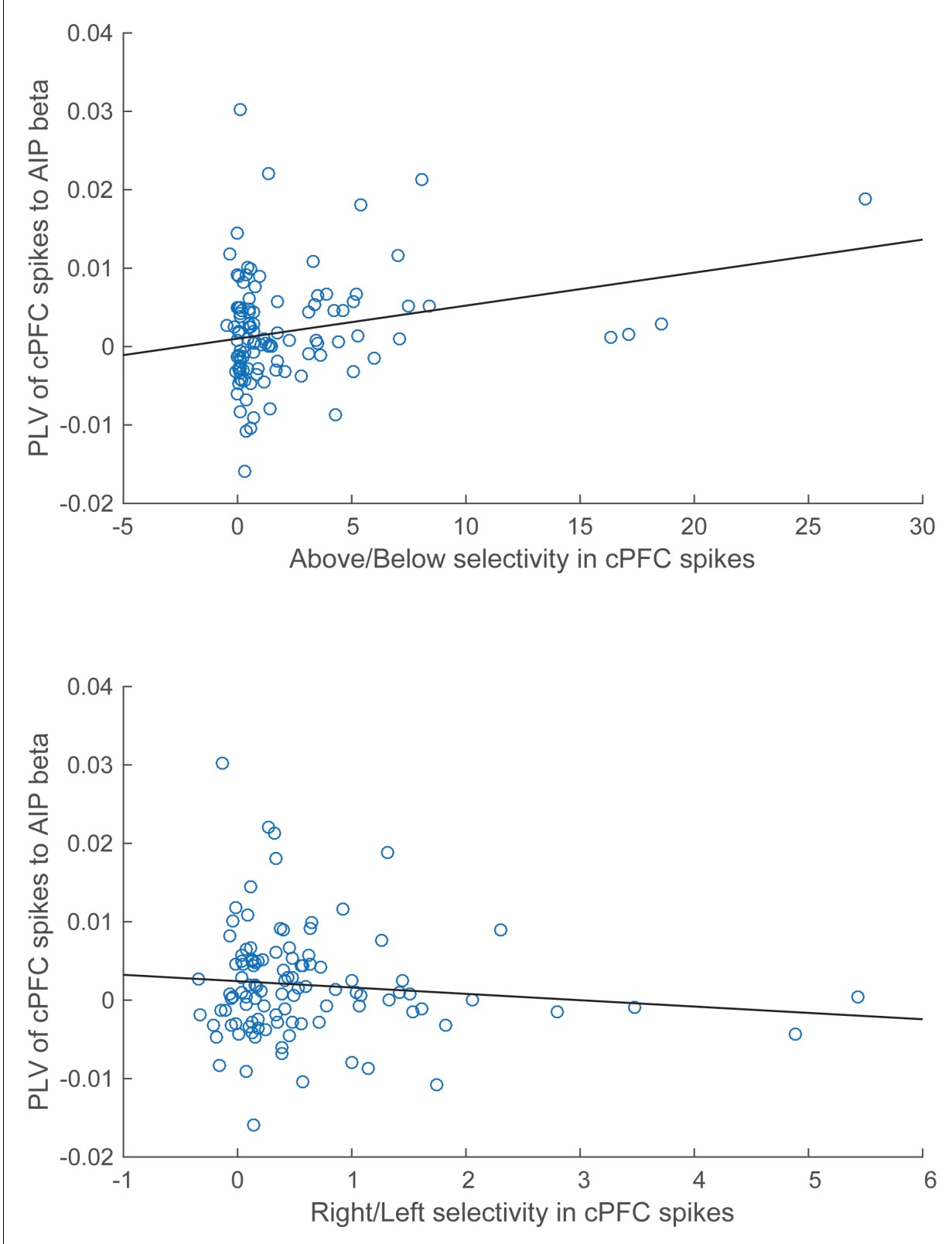

**Figure 9.** Correlation of spike-LFP synchrony with spiking selectivity. Synchrony of cPFC spikes to AIP LFP beta-band oscillations, during the Category epoch. Above/Below selectivity in spiking (top) is positively correlated with the spike synchrony (averaged across all their electrode pairings) to parietal beta rhythms (r=0.25, p=0.009), whereas Right/Left selectivity (bottom) is not (r=−0.11, p=0.28).

## Task design

Neural data were collected from 39 experiments (22 experiments from animal G and 17 from animal P). Experimental control was implemented via Cortex (NIMH, Laboratory of Neuropsychology) and infrared eye-tracking, sampled at 500 Hz, via Eyelink 1000 (SR Research Ltd, Mississauga, Canada). The animals were seated in customized primate chairs and placed inside sound-attenuating chambers. Visual stimuli were presented at full contrast on a CRT monitor (at a distance of approx. 50 cm from the animal), refreshing at 100 Hz.

The task design is shown in *Figure 1A*. Trials began when the animal held a bar and maintained visual fixation on a central target for 1 s. The fixation target was a red square (0.25-degree side), flanked on the right and left side with 2 white horizontal lines of 0.2-degree thickness and 6-degree length. These lines, each of which had a 2-degree gap from the fixation target (the allowable fixation window had a 1.5-degree radius), were only displayed for the 1 s duration of the fixation window, in order to help the animals re-calibrate the division between upper/lower visual hemifields after the previous trial. The sample stimulus was displayed on the screen 200 ms after the lines had been turned off (the fixation target stayed on the screen for the duration of the trial and the animals were required to maintain fixation throughout). Sample stimuli were circular white dots (0.2-degree radius), the location of which was selected randomly from 16 possible locations (four in each visual quadrant), plus a random jitter of ±0.5 degree on each axis. Overall, the sample could appear in any one of 144 distinct locations, spanning an area of 170 degrees$^2$. No samples were displayed inside the animals' spatial window of fixation. Sample display lasted for 800 ms and was followed by a 500 ms delay interval, during which only the fixation target appeared on the screen. Up to this point of the trial the animals were required to determine whether the sample had appeared in the upper or lower visual hemifield (i.e., category Above or Below) and hold the sample category in working memory. At the end of the delay interval, the two white horizontal lines (hemi-boundaries) re-appeared at new locations. A clockwise (CW) or counter-clockwise (CCW) shift was randomly selected on each trial. A CW shift meant that the left hemi-boundary appeared 4 degrees above the horizontal meridian and the right hemi-boundary appeared 4 degrees below the horizontal meridian. A CCW shift was implemented in the reverse way. During the 1 s interval following the new boundary display, the animals were required to adjust the category decision criterion. Note that the new location of the boundaries *did not change the sample category.* Rather, it only changed the *spatial area* that corresponded to that category. After the boundary shift, the new retinotopic spatial area of the Above/Below categories overlapped with the pre-shift retinotopic area by only 50%. Because of this, test stimuli that were very close to the sample location could potentially be a non-match to the sample category, and test stimuli far away from the sample location could potentially be a match to the sample category.

At the end of this delay interval, a test stimulus, identical in form to the sample, was displayed for 1 s. The location of the test stimulus was selected randomly from one of the 624 locations shown in *Figure 1*B. If the test stimulus matched the category of the sample stimulus (Above or Below the hemi-boundaries), the animals had to release the bar before the end of its display for liquid reward. If it did not match the sample category, the animals were required to maintain contact with the bar and visual fixation for another 0.5 s following the test, when a second test would appear on the screen. The second test would always be a match and the animals would release the bar for reward.

## Neural recordings and data analyses

All neurophysiological recordings were performed with the Cerebus Neural Processing System (Blackrock Microsystems, Salt Lake City, Utah). Simultaneous multi-electrode recordings were made from the left frontal cortex and the left posterior parietal cortex, using two custom-made multi-electrode arrays (each array with 8–25 tungsten electrodes; FHC, Bowdoin, ME). Electrodes from both arrays were acutely lowered every day of recording, either individually or in pairs, at depths that were guided both by the MRI images (to target the areas of interest) and the success in isolating neural spiking activity. This method of multi-electrode/multi-area recording allowed us to: (a) sample neural signals from relatively large surface areas (each array covered up to 250 mm$^2$, with electrodes that could be no closer than 1 mm from each other), (b) sample neural signals from different sets of locations every day of recording, (c) evaluate the real-time cross-area dynamic interactions in the

frequency and time domains, and (d) make cross-area comparisons of neural processing under identical experimental conditions.

Neural data from the posterior parietal cortex came from the rostral part of the lateral bank of the intraparietal sulcus (anterior intraparietal area; AIP). Our analyses of prefrontal neural data distinguished between the regions of the principal sulcus (lateral PFC; lPFC) and the more posterior area 8A (caudal PFC; cPFC), which lies at the junction between the prefrontal and premotor cortices. Area 8A, extending from the caudal end of the principal sulcus up to the arcuate sulcus, also includes the frontal eye field (FEF). Results from the FEF in isolation were similar to the results from the rest of area 8A, and were pooled together in the cPFC group. In addition, results from area 45 were similar to those from area 46 and pooled together in the lPFC group. Additional recordings were made from various adjacent frontal and parietal areas, but in this report we are focusing on the three areas that gave us the bulk of the data from both monkeys.

Electrode recordings were first fed to a unity-gain headstage and were referenced to ground. The recorded signals were band-pass filtered to separate spikes from LFPs (0.3–500 Hz for LFPs and 250–7500 Hz for spikes). LFPs were digitized at 1 KHz sampling rate and spikes were digitized at 30 KHz. To ensure that only signals from active regions of the brain were collected, LFPs were recorded only from sites that also displayed spiking activity.

Spikes were sorted into clusters of putative single-neuron activity using Offline Sorter v3 (Plexon; Dallas, TX), based on principal component analysis. Firing rates (in spikes/second) were computed from spike counts over contiguous 20 ms bins and convolved with a 200 ms Gaussian window. For selectivity analyses, the firing rates were z-transformed based on the mean and variance of firing rate during a 3 s pre-trial baseline. Neural selectivity of spiking activity was quantified in Matlab as the bias-corrected, omega-variant of percent explained variance (ωPEV; *Brincat and Miller, 2015*; *Buschman et al., 2011*).

LFPs were bi-directionally bandstop filtered (59–61 Hz), using a 10th order Butterworth filter, to minimize the component of 60-cycle noise. The LFPs were subsequently downsampled at 333 Hz, and centered on the cross-trial mean. The latter processing step reduced the amplitude of stimulus-evoked potentials and helped isolate the trial-induced oscillations. The LFPs were then decomposed to their spectral components using a Matlab-based wavelet analysis toolbox (*Torrence and Compo, 1998*; offered at the URL: http://atoc.colorado.edu/research/wavelets/), through their convolution with a Morlet wavelet, at six octaves (from 2 Hz to 128 Hz) at a 0.1- octave resolution. Quantification of LFP-LFP synchrony (pairwise phase consistency) relied on the Matlab-based toolbox FieldTrip (*Oostenveld et al., 2011*). Finally, spike-LFP synchrony was quantified as the single-trial phase-locking value (PLV), which measures the angular concentration of the frequency-specific instantaneous phases of the LFP at the time of spikes (*Antzoulatos and Miller, 2014*; *Siegel et al., 2009*). For illustration purposes, the group-averaged data were smoothed with kernels of FWHM less than 0.5 octave and/or 100 ms. All statistical analyses were performed on unsmoothed data using Matlab, and corrections for multiple comparisons followed the Bonferroni method. Unless noted, group data are reported as average ± SEM.

## Acknowledgements

The authors wish to thank PC Alexander, D Fioravante, C Ranganath, WM Usrey, and DK Warland for helpful discussions, A Wutz for comments on the manuscript, R Norton for IT management, and BC Gray for technical assistance. This work was supported by NIMH 4R01MH065252 (EKM) and Prop. 63 the Mental Health Services Act and the Behavioral Health Center of Excellence at UC Davis (EGA).

## Additional information

### Funding

| Funder | Grant reference number | Author |
| --- | --- | --- |
| University of California, Davis | Behavioral Health Center of Excellence, Pilot Award | Evan G Antzoulatos |
| Mental Health Services Act, | Pilot Award | Evan G Antzoulatos |

Proposition 63

| | | |
|---|---|---|
| National Institute of Mental Health | R01MH065252 | Earl K Miller |

The funders had no role in study design, data collection and interpretation, or the decision to submit the work for publication.

## Author contributions

EGA, Conception and design, Acquisition of data, Analysis and interpretation of data, Drafting or revising the article; EKM, Conception and design, Analysis and interpretation of data, Drafting or revising the article

## Ethics

Animal experimentation: All work was in accordance with the National Institutes of Health guidelines and approved by the Massachusetts Institute of Technology Committee for Animal Care (protocol number: 0516-026-19).

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
