## [Decision Letter]

Thank you for submitting your article "Synchronous β rhythms of frontoparietal networks support only behaviorally relevant representations" for consideration by *eLife*. Your article has been reviewed by two peer reviewers, and the evaluation has been overseen by Tatiana Pasternak as the Reviewing Editor and Timothy Behrens as the Senior Editor. The following individuals involved in review of your submission have agreed to reveal their identity: Thilo Womelsdorf (Reviewer #1) and Matthew Chafee (Reviewer #2).

The reviewers have discussed the reviews with one another and the Reviewing Editor has drafted this decision to help you prepare a revised submission.

Summary statement:

This study examines long-range frontoparietal interactions by evaluating activity recorded simultaneously from prefrontal and posterior parietal cortex during a delayed match-to-spatial category task. The authors recorded both spiking activity and the LFPs and revealed category-selective synchrony of β-band oscillations that were selective for task-relevant information. Although category-selective frontoparietal synchrony was also observed in the δ band, this synchrony was present for both task-relevant and the task-irrelevant information. The data suggest that while spiking activity in the prefrontal cortex showed selectivity for both relevant and irrelevant information, only neurons with task relevant information were synchronized with the parietal β rhythms. The authors conclude that the synchrony of long-range β rhythms may support only the representation of task variables that are relevant to the category decision.

Both reviewers were impressed with the work, listing a number of strengths, including the design of the behavioral task, approaches to data analysis, careful statistics and clarity of writing. However, they both raised a number of issues that should be addressed in the revision. These are listed below:

*Reviewer 1:*

1) Provide information about the baseline used for each analysis to characterize enhanced β synchronization.

2) Extend the analysis of the top 10% informative firing rates to neurons with no task-relevant information. The reviewer suggests that adding a scatterplot showing category specific information carried in spiking activity against the category information in spike-LFP with strengthen the conclusion that only informative spiking activity is synchronized to the parietal β rhythms.

3) Address the question of the relationship between LFP sites with category selectivity (left/right; above/below) and right/left spiking spatial selectivity.

4) Clarify the statement in the second paragraph of the subsection “Directional frontoparietal interactions”, that LFP pairs with the top 10% category selective β coherence were used for spike-LFP analysis. The referee is concerned about the selection bias and asks whether selecting 10% of spike-LFP pairs with β synchrony would also show strong category selective coherence.

5) Provide information about the% of explained variance carried by neurons in the top 10% neurons with category distinction.

6) Address the potentially misleading label "Directional frontoparietal interactions".

7) Clarify frequency bins in Figure 2.

*Reviewer 2:*

1) Address the comment about δ synchrony and the possibility that synchrony in this band may not be a reflection of the motor response.

2) The reviewer points out that the analysis involving subtracting trial-average LFP may not reveal bottom-up oscillatory components of higher frequencies phase-locked to stimulus onset.

3) Address the relationship between category selective phase coherence between two sites and category preference of each of these sites.

4) Address whether the representation of categories "below/above" is balanced. This reviewer suggests separating Figure 4 into "above" and "below" categories to assess whether these categories are supported by distinct network dynamics.

5) Please address the problem raised by this reviewer with respect to Figure 7 and Figure 8.

This reviewer also made a number of useful suggestions, including the addition of labels in Figure 2, adding grayscale panels in Figure 1, providing the rational for selecting AIP as the recording site, explaining the reason for using both phase coherence and pairwise phase consistency.

[Editors' note: further revisions were requested prior to acceptance, as described below.]

Thank you for resubmitting your work entitled "Synchronous β rhythms of frontoparietal networks support only behaviorally relevant representations" for further consideration at *eLife*. Your revised article has been favorably evaluated by Timothy Behrens (Senior editor), a Reviewing editor, and two reviewers.

The manuscript has been improved but there are some remaining issues that need to be addressed before acceptance, as outlined below:

1) Please address the question from Reviewer 1 whether the β effects represent enhancement vs suppression of activity relative to the prestimulus baseline levels. Also, respond to the comment concerning the period at which firing rate and LFP selectivity start to correlate.

2) Reviewer 2 had problems identifying the relevant figures (see below). Please clarify. The problem may be the consequence of the individual figures included in the revision were not numbered. Please submit a version with appropriately labeled figures.

*Reviewer #1:*

These are very strong revisions. They enhance the manuscripts conclusions. There were, however, two aspects that I want to raise.

1) There was some misunderstanding regarding reviewer 1 – comment 1:

The current text does not allow to discern for how many LFP sites, LFP-LFP pairs and spike-LFP pairs are the β effects in the category- and shift- period is reflecting an enhancement or a suppression relative to a prestimulus baseline period. The question such an analysis should answer is whether the selectivity for above or below categories is based to a significant extent based on a selective decrease in power/coherence for one category as opposed to being due to a selective increase for one category.

The current text conveys that power coherence selectively increases for one category (above or below) but the data seem ambiguous in that the selectivity can also originate in a selective loss of power/coherence for one task category relative to a baseline (or intertrial power/coherence) level.

2) The newly added correlation results (Figure 8) are very interesting. Can the authors add the information on whether Pearson and Spearman rank correlations provided similar results? These new findings sort of call upon a time resolved analysis to identify when in time rate and LFP selectivities start to correlate. Doing such an analysis would enhance the current manuscript's appeal to discern mechanistic explanations, but I leave it to the authors to consider this aspect maybe for a future study.

*Reviewer #2:*

The authors have thoroughly addressed my prior comments.

I was unable to find the new Figure 2—figure supplement 3 plotting synchrony separately for match and nonmatch trials (to demonstrate relation to response). The 4th supplemental item shows instead synchrony broken out ABOVE/BELOW and RIGHT/LEFT categories.

---

## [Author Response]

*[…] Reviewer 1:*

1) Provide information about the baseline used for each analysis to characterize enhanced β synchronization.

In the revised manuscript (subsection “Predominant beta rhythms in local and long-range oscillations of the frontoparietal network”, fifth paragraph), we included additional information to clarify how we created surrogate datasets to estimate the baseline, chance-level synchrony in our analyses.

*2) Extend the analysis of the top 10% informative firing rates to neurons with no task-relevant information. The reviewer suggests that adding a scatterplot showing category specific information carried in spiking activity against the category information in spike-LFP with strengthen the conclusion that only informative spiking activity is synchronized to the parietal β rhythms.*

We have now included Figure 8—figure supplement 4, which contrasts PLV for Above/Below selectivity vs. non-selectivity. This further illustrates that spikes with Above/Below selectivity are more tightly coupled to parietal β rhythms than spikes with no selectivity. We have also added a scatterplot (Figure 9) of PLV vs. spiking selectivity. This also reinforces the conclusion that the strength of Above/Below selectivity correlates with the parietal β synchrony.

*3) Address the question of the relationship between LFP sites with category selectivity (left/right; above/below) and right/left spiking spatial selectivity.*

This comment is presumably related to the second reviewer’s comment #3, regarding the relationship between a site’s category preference and the category-selectivity of synchrony between sites. Per the reviewer’s suggestion, we examined the relationship between each electrode’s spiking selectivity and the selectivity of LFP synchrony of the corresponding electrode’s pairings with other electrodes. We now plot the category-selective LFP-LFP synchrony separately for pairs of electrodes with the same category preference vs. electrodes with different preferences. We do this for both Above/Below and for Right/Left categories (Figure 6—figure supplement 2). This revealed that the dynamics of category-selective synchrony are similar between electrodes with the same preference to electrodes with different preferences.

*4) Clarify the statement in the second paragraph of the subsection “Directional frontoparietal interactions”, that LFP pairs with the top 10% category selective β coherence were used for spike-LFP analysis. The referee is concerned about the selection bias and asks whether selecting 10% of spike-LFP pairs with β synchrony would also show strong category selective coherence.*

After re-reading that statement, we realized that it could be confusing and we re-wrote it in a more straightforward way (subsection “Asymmetric frontoparietal interactions”, second paragraph). To address the concern about the selection bias, we followed the reviewer’s suggestion and added a new figure (Figure 7—figure supplement 3). It separately illustrates the category-selective synchrony of frontoparietal LFPs for electrodes with and without spike-LFP synchrony. As the figure illustrates, there is pronounced category selectivity of β rhythms between the electrodes that also have strong spike-LFP synchrony.

*5) Provide information about the% of explained variance carried by neurons in the top 10% neurons with category distinction.*

This is now supplied in a new table, Figure 8—figure supplement 3.

6) Address the potentially misleading label "Directional frontoparietal interactions".

In order to avoid any potential misconception from our readers, we have now renamed that section to “Asymmetric frontoparietal interactions”. It is technically more accurate. We edited the text accordingly.

*7) Clarify frequency bins in Figure 2.*

This is now clarified in the main text (subsection “Predominant beta rhythms in local and long-range oscillations of the frontoparietal network”) and the legend of Figure 2. The frequency bins were one tenth of the corresponding octave.

*Reviewer 2:*

*1) Address the comment about δ synchrony and the possibility that synchrony in this band may not be a reflection of the motor response.*

We have included a new figure (Figure 2—figure supplement 2). It plots the synchrony separately for match (early motor response) vs. non-match (late motor response) trials. It indicates strong δ synchrony at the time of the motor response. The main focus of our manuscript is on β but we welcome the opportunity to clarify the role of the δ band. We discuss how strong δ-band synchrony during the test epoch may be related to the motor response, the oculomotor response that follows release from fixation, or the reward (subsection “Predominant beta rhythms in local and long-range oscillations of the frontoparietal network”, last paragraph and legend of Figure 2—figure supplement 2).

*2) The reviewer points out that the analysis involving subtracting trial-average LFP may not reveal bottom-up oscillatory components of higher frequencies phase-locked to stimulus onset.*

That is true. We point this out but also discuss that removing the trial-average is imperative to avoid any spurious synchrony that may arise from parallel stimulus-evoked changes in the LFP signals (subsection “Predominant beta rhythms in local and long-range oscillations of the frontoparietal network”, first paragraph). We also note that it is a common practice in the extant literature for this very reason. But to fully inform the reader and to address the reviewer’s concern, we include a new figure (Figure 2—figure supplement 3) showing the power and synchrony of LFPs without the removal of trial average, which does not show effects on the high frequencies.

*3) Address the relationship between category selective phase coherence between two sites and category preference of each of these sites.*

To address this comment as well as the 1^st^ reviewer’s comment #3, we now provide a plot of the category selectivity of LFP synchrony separately for pairs of electrodes with the same category preference in their spike rate vs. electrodes with different preferences (Figure 6—figure supplement 2). Synchrony of LFP signals is similar between electrodes with common vs. different category preferences.

*4) Address whether the representation of categories "below/above" is balanced. This reviewer suggests separating Figure 4 into "above" and "below" categories to assess whether these categories are supported by distinct network dynamics.*

Per the reviewer’s suggestion, we have now included a new figure (Figure 4—figure supplement 2). It illustrates the time-frequency dynamics of synchrony separately for the Above and Below categories. These dynamics were similar for both categories and for both between- and within-area synchrony.

*5) Please address the problem raised by this reviewer with respect to Figure 7 and Figure 8.*

The issue in reference here presumably refers to the comment above (#4). We have added new supplemental figures (Figure 7—figure supplement 1 and Figure 8—figure supplement 2), which illustrate the corresponding spike-field PLV results separately for categories Above and Below.

*This reviewer also made a number of useful suggestions, including the addition of labels in Figure 2, adding grayscale panels in Figure 1, providing the rational for selecting AIP as the recording site, explaining the reason for using both phase coherence and pairwise phase consistency.*

Per the reviewer’s suggestion, we have added new panels in the revised version of Figure 1 (namely the probability of classification as Below and Right) and more labels in Figure 2 (including an indication of the β band used in our analyses). We also explain that we focused on AIP because it is an area known to be tightly connected to premotor areas and be involved in visuospatial transformations that guide motor movements (subsection “Predominant beta rhythms in local and long-range oscillations of the frontoparietal network”, first paragraph). Pairwise phase consistency and coherence are related measures that yield similar results as long as the number of trials is large (which they are). Thus, to avoid any possible confusion and for the sake of brevity, we now only report pairwise phase consistency results in the revised manuscript.

[Editors' note: further revisions were requested prior to acceptance, as described below.]

*[…] Reviewer #1:*

*These are very strong revisions. They enhance the manuscripts conclusions. There were, however, two aspects that I want to raise.*

*1) There was some misunderstanding regarding reviewer 1 – comment 1:*

*The current text does not allow to discern for how many LFP sites, LFP-LFP pairs and spike-LFP pairs are the β effects in the category- and shift- period is reflecting an enhancement or a suppression relative to a prestimulus baseline period. The question such an analysis should answer is whether the selectivity for above or below categories is based to a significant extent based on a selective decrease in power/coherence for one category as opposed to being due to a selective increase for one category.*

*The current text conveys that power coherence selectively increases for one category (above or below) but the data seem ambiguous in that the selectivity can also originate in a selective loss of power/coherence for one task category relative to a baseline (or intertrial power/coherence) level.*

We thank the reviewer for the very helpful clarifications. Because the question is whether category selectivity arises from selective loss or gain for the preferred/non-preferred categories, we addressed it in the context of synchrony between LFPs (in our manuscript we do not report category selectivity in the LFP power or in the spike-LFP synchrony). In the revised manuscript, we have added a new figure (Figure 4—figure supplement 3), which compares the PPC values for the preferred and non-preferred categories to those of the pre-trial fixation epoch.

We did this analysis separately for the β-band and δ-band synchrony of the Category epoch and of the Shift epoch. These new results appear in the fourth paragraph of the subsection “Category-selective synchrony”. The β synchrony of the Category epoch (when its selectivity was stronger) increases for the preferred category and decreases for the non-preferred category. Similarly, the δ synchrony during the Shift epoch (when its selectivity was stronger) increases for the preferred category and decreases for the non-preferred category. In the alternative epochs (when the corresponding band-specific PPCs were weaker) there is a decrease from baseline, but to a smaller extent for the preferred than the non-preferred categories.

*2) The newly added correlation results (Figure 8) are very interesting. Can the authors add the information on whether Pearson and Spearman rank correlations provided similar results? These new findings sort of call upon a time resolved analysis to identify when in time rate and LFP selectivities start to correlate. Doing such an analysis would enhance the current manuscript's appeal to discern mechanistic explanations, but I leave it to the authors to consider this aspect maybe for a future study.*

The linear and rank correlation coefficients show a similar trend, and neither of them is significant for the correlation between Right/Left spiking selectivity and spike-LFP synchrony. However, whereas the linear coefficient indicates significant correlation between Above/Below spiking selectivity and spike-LFP synchrony, the rank coefficient does not reach statistical significance. In the revised manuscript (Results, last paragraph) we now report both Pearson’s linear and Spearman’s rank correlation coefficients.

The comment on the time-resolved correlation is very interesting, and we intend to address it in the near future. Please note that the observed correlation applies to the entire trial epoch, and it may not be present at any individual time point. For instance, spiking selectivity at time X may not correlate with synchrony at time X, but it may do so at time X+500 ms, still within the same trial epoch. In order to be as thorough with this analysis as we have been with all other analyses we report in this manuscript, we should analyze *time-resolved cross-correlation,* which would be computationally daunting and very time-consuming, given the size of our dataset. We think this analysis would more appropriate for another report, which would focus on detailed mechanistic analyses of some of the phenomena we identify in this manuscript. We thank the reviewer for suggesting this analysis to us and for not requiring it for acceptance of the manuscript.

*Reviewer #2:*

*The authors have thoroughly addressed my prior comments.*

I was unable to find the new Figure 2—figure supplement 2 plotting synchrony separately for match and nonmatch trials (to demonstrate relation to response). The 4th supplemental item shows instead synchrony broken out ABOVE/BELOW and RIGHT/LEFT categories.

The reviewer is absolutely justified not finding our Figure 2—figure supplement 2, as we inadvertently omitted it when we uploaded the material to the *eLife* website. We should have caught this omission upon proofreading the merged manuscript, but we didn’t, and we apologize for that. We have now uploaded all 24 figures. Furthermore, we have added a label inside each figure, indicating the corresponding figure number.